# Tension-driven multi-scale self-organisation in human iPSC-derived muscle fibers

Qiyan Mao[1]*[†], Achyuth Acharya[1†], Alejandra Rodríguez-delaRosa[2,3,4], Fabio Marchiano[1], Benoit Dehapiot[1], Ziad Al Tanoury[2,3,4], Jyoti Rao[2,3,4], Margarete Díaz-Cuadros[2,3,4], Arian Mansur[4,5], Erica Wagner[2], Claire Chardes[1], Vandana Gupta[5], Pierre-François Lenne[1], Bianca H Habermann[1], Olivier Theodoly[6], Olivier Pourquié[2,3,4]*, Frank Schnorrer[1]*

[1]Aix Marseille University, CNRS, IBDM, Turing Centre for Living Systems, Marseille, France; [2]Department of Pathology, Brigham and Women's Hospital, Boston, United States; [3]Department of Genetics, Harvard Medical School, Boston, United States; [4]Harvard Stem Cell Institute, Boston, United States; [5]Division of Genetics, Department of Medicine, Brigham and Women's Hospital, Harvard Medical School, Boston, United States; [6]Aix Marseille University, CNRS, LAI, Turing Centre for Living Systems, Marseille, France

*For correspondence:
qiyan.mao@univ-amu.fr (QM);
pourquie@genetics.med.harvard.edu (OP);
frank.schnorrer@univ-amu.fr (FS)

[†]These authors contributed equally to this work

Competing interest: The authors declare that no competing interests exist.

**Abstract** Human muscle is a hierarchically organised tissue with its contractile cells called myofibers packed into large myofiber bundles. Each myofiber contains periodic myofibrils built by hundreds of contractile sarcomeres that generate large mechanical forces. To better understand the mechanisms that coordinate human muscle morphogenesis from tissue to molecular scales, we adopted a simple in vitro system using induced pluripotent stem cell-derived human myogenic precursors. When grown on an unrestricted two-dimensional substrate, developing myofibers spontaneously align and self-organise into higher-order myofiber bundles, which grow and consolidate to stable sizes. Following a transcriptional boost of sarcomeric components, myofibrils assemble into chains of periodic sarcomeres that emerge across the entire myofiber. More efficient myofiber bundling accelerates the speed of sarcomerogenesis suggesting that tension generated by bundling promotes sarcomerogenesis. We tested this hypothesis by directly probing tension and found that tension build-up precedes sarcomere assembly and increases within each assembling myofibril. Furthermore, we found that myofiber ends stably attach to other myofibers using integrin-based attachments and thus myofiber bundling coincides with stable myofiber bundle attachment in vitro. A failure in stable myofiber attachment results in a collapse of the myofibrils. Overall, our results strongly suggest that mechanical tension across sarcomeric components as well as between differentiating myofibers is key to coordinate the multi-scale self-organisation of muscle morphogenesis.

## Editor's evaluation

This manuscript describes pioneering work providing a detailed description of iPS-derived muscle fiber differentiation in culture. It demonstrates that muscle fibers show self-organising capacities in vitro and form bundles with identified attachment points; this self-organisation generates internal tension within myofibers. Overall, this study suggests that tension drives sarcomerogenesis in multi-fibrillar vertebrate muscles and will be of interest to researchers in the muscle field and also biophysicists interested in collective cell behaviour.

## Introduction

Human skeletal muscles display a very hierarchical and stereotypical architecture. At its distal ends, each muscle is mechanically connected to different skeletal elements, which are moved by the contracting muscle to achieve locomotion. Within each muscle, contractile muscle cells called myofibers are organised into myofiber bundles, several of which are combined to form one human muscle (*Hill and Olson, 2012*). Within each muscle fiber, the contractile myofibrils are laterally aligned displaying the well-appreciated cross-striated morphology of skeletal muscle fibers (*Huxley and Niedergerke, 1954*; *Huxley and Hanson, 1954*). Each myofibril consists of a long chain of contractile machines called sarcomeres, which extend across the entire length of the muscle fiber to mechanically connect the attached skeletal elements (*Schiaffino and Reggiani, 2011*). As such, coordinated contraction of all sarcomeres shortens all myofibrils and thus shortens each muscle fiber. This results in force production powering the desired body movement (*Lemke and Schnorrer, 2017*; *Lindstedt, 2016*).

Sarcomere architecture is highly regular. The plus ends of parallel actin filaments are cross-linked at the Z-discs bordering each sarcomere, while their minus ends point centrally where they overlap with large bipolar myosin filaments. The latter are cross-linked in the middle of the sarcomere building the M-band (*Dasbiswas et al., 2018*; *Lange et al., 2006*). Both types of filaments are stably linked together by the gigantic elastic protein titin, whose N-terminus is bound to α-actinin at the Z-disc, while its C-terminus is embedded into the M-band (*Lange et al., 2006*; *Linke, 2018*; *Tskhovrebova and Trinick, 2003*). This results in a stereotypic sarcomere length, depending on muscle type ranging from 2 to 3 µm in relaxed human muscle in vivo (*Ehler and Gautel, 2008*; *Llewellyn et al., 2008*; *Tskhovrebova and Trinick, 2003*).

As most mature human muscles are several centimetres long, many thousand sarcomeres need to be concatenated into each myofibril to mechanically connect both myofiber ends (*Lemke and Schnorrer, 2017*). Assembling the correct number of sarcomeres will result in the optimal amount of passive tension present in the relaxed muscle. This is a prerequisite for effective active muscle force production during sarcomere contraction, powered by myosin forces pulling on actin (*Dasbiswas et al., 2018*; *Linke, 2018*). Hence, the complex hierarchical muscle tissue operates under continuous mechanical tension.

Optimal tension levels are not only important in the mature muscle, but also essential during its development. Insights from in vivo imaging of myofibrillogenesis during *Drosophila* flight muscle development sparked the tension-driven self-organisation hypothesis of myofibrillogenesis (*Weitkunat et al., 2014*): this hypothesis suggests that myotubes first attach to tendons. After successful attachment, they build up mechanical tension along the long axis of the muscle fiber. This tension build-up triggers the simultaneous assembly of long periodic myofibrils that extend through the entire fiber and thus mechanically connect the two attached exoskeletal elements of the fly (*Weitkunat et al., 2014*; *Weitkunat et al., 2017*). Thus, tension is key to coordinate myofibril self-organisation over long distances in insect muscles.

However, *Drosophila* muscles are simple, single myofiber muscles that do not form higher-order myofiber bundles. Hence, it is a limited model for understanding how myofiber attachment, myofiber bundling, and sarcomere assembly are coordinated during human muscle tissue morphogenesis. Furthermore, human muscle tissue not only connects to tendon and cartilage cells, which are the source of various guidance cues in vivo (*Schweitzer et al., 2010*), but also contains a complex muscle-associated connective tissue that may further instruct muscle development both mechanically and chemically (*Helmbacher and Stricker, 2020*). How precisely this complex interplay of cell types may coordinate muscle morphogenesis at different scales is not well understood.

Here, we set out to test the self-organisation capacity of human muscle tissue in vitro when developing on a substrate lacking any localised guidance, attachment, or other mechanical cues. We employed a recently developed human induced pluripotent stem cell (iPSC) differentiation protocol that can efficiently generate myogenic precursors which further differentiate robustly into striated skeletal muscle fibers in vitro (*Chal et al., 2015*; *Chal et al., 2016*; *Al Tanoury et al., 2021*; *Yan et al., 2021*). This method allowed us to obtain large quantities of isogenic human myogenic precursors of defined developmental history, in contrast to primary myoblast cultures from donors. Furthermore, CRISPR-based genome engineering enabled us to tag sarcomeric protein components of choice to follow their live dynamics in vitro (*Sharma et al., 2018*; *Al Tanoury et al., 2021*).

Using two-dimensional (2D) cultures of human iPSC-derived myogenic precursors, we found that despite the absence of attachment cues, myofibers elongate to self-restricted stable lengths. Concomitantly, they self-organise into large myofiber bundles that attach to shared attachment foci at both ends. As myofibers approach stable lengths, their sarcomeres emerge simultaneously over broad regions, suggesting that myofibril and myofiber morphogenesis are coordinated from tissue to molecular scales. We identified mechanical tension as a likely coordination mechanism as tension precedes sarcomere formation and further increases when myofibrils mature. Consequently, integrin-based adhesions between the bundled myofiber ends support stable myofiber attachment. Interestingly, faster bundling results in accelerated sarcomere assembly. Since tension can be transmitted across tissue, cellular, and molecular scales, we hypothesise that tension is key to coordinate muscle morphogenesis across scales.

## Results

### Human iPSC-derived myofibers follow a biphasic differentiation process

In order to study human muscle fiber morphogenesis on cellular and subcellular scales, we adapted a 2D in vitro culture system. We generated human iPSC-derived myogenic precursors by differentiating NCRM1 iPS cells for 20–30 days according to *Chal et al., 2016*; *Al Tanoury et al., 2021*. These cultures were then dissociated and filtered through 70 µm and then 40 µm cell filters to obtain single-cell suspensions. These suspensions were replated in SKGM-2 medium for 1–2 days to generate cultures enriched in myogenic precursors prior to being frozen (*Chal et al., 2016*; *Al Tanoury et al., 2021*). Frozen myogenic precursors were thawed, seeded on Matrigel-coated ibidi glass-bottom dishes and grown in proliferation medium (SKGM-2) for 2–3 days before being switched to KCTiP differentiation medium (see Materials and methods). On day 1 in KCTiP differentiation medium, the myogenic precursors initially resemble fibroblasts by shape (*Figure 1A*). These myogenic precursors are able to fuse to other myogenic precursors or elongated myotubes to form multi-nucleated myotubes (*Figure 1—video 1*, *Figure 1—figure supplement 1A*).

Concomitant with myogenic precursors fusion, myotubes rapidly elongate, while their width decreases during the first week (*Figure 1A–F, A'–F', J and K*, *Figure 1—video 1*, *Figure 1—figure supplement 1B*), suggesting that they are actively extending to increase their aspect ratio. The average length of a day 7 myotube/myofiber is about 1.2 mm, while being only 3–5 µm in width in our culture conditions. During the second week of culture, average myofiber length stabilises at about 1.2–1.3 mm (*Figure 1G–K*), while fusion continues, reaching an average of 15 nuclei per myofiber at day 15 (*Figure 1L*). This happens despite that fact that our culture dishes have an average dimension of 6 mm, suggesting that a self-limiting mechanism may prevent further elongation of the myofibers. We conclude that on our unpatterned 2D substrates, differentiating myotubes/myofibers follow a biphasic behaviour of elongation until day 7, followed by a stabilisation in length until day 15.

### Myofibers self-organise into bundles that merge to stable sizes

To better understand how myofibers reach stable lengths at day 7, we investigated their spatial organisation with respect to neighbouring myofibers. We found that from days 3 and 4, myotubes and myofibers generate swirly patterns that enlarge over time (*Figure 2A–F*), indicating that new myofibers are aligning with growing myofiber bundles despite the lack of predefined orientation cues. This suggests a self-organisation mechanism during myofiber bundling.

To quantitatively analyse the alignment of myotubes and myofibers with their neighbours, we first displayed the local fiber orientation as vector fields for the central region of the culture (*Figure 2—figure supplement 1*, see Materials and methods). At day 3 we observed small swirly patterns in vector fields (*Figure 2—figure supplement 1A*), indicative of short-range alignments of polarity. These aligned regions enlarge over time (*Figure 2—figure supplement 1A-F*), which is supported by an increasing average nematic length during differentiation (*Figure 2—figure supplement 1G, H*; nematic length was defined as the intersection of the initial linear decay of the spatial correlation function with the x axis, see Materials and methods). To visualise the local discontinuities of alignment, which often locate at the end of myofiber bundles, we calculated the nematic order for each vector grid (*Figure 2—figure supplement 1A'-F'*; nematic order is a parameter describing the alignment of local orientation of the vector field at a given position, see Materials and methods). This analysis

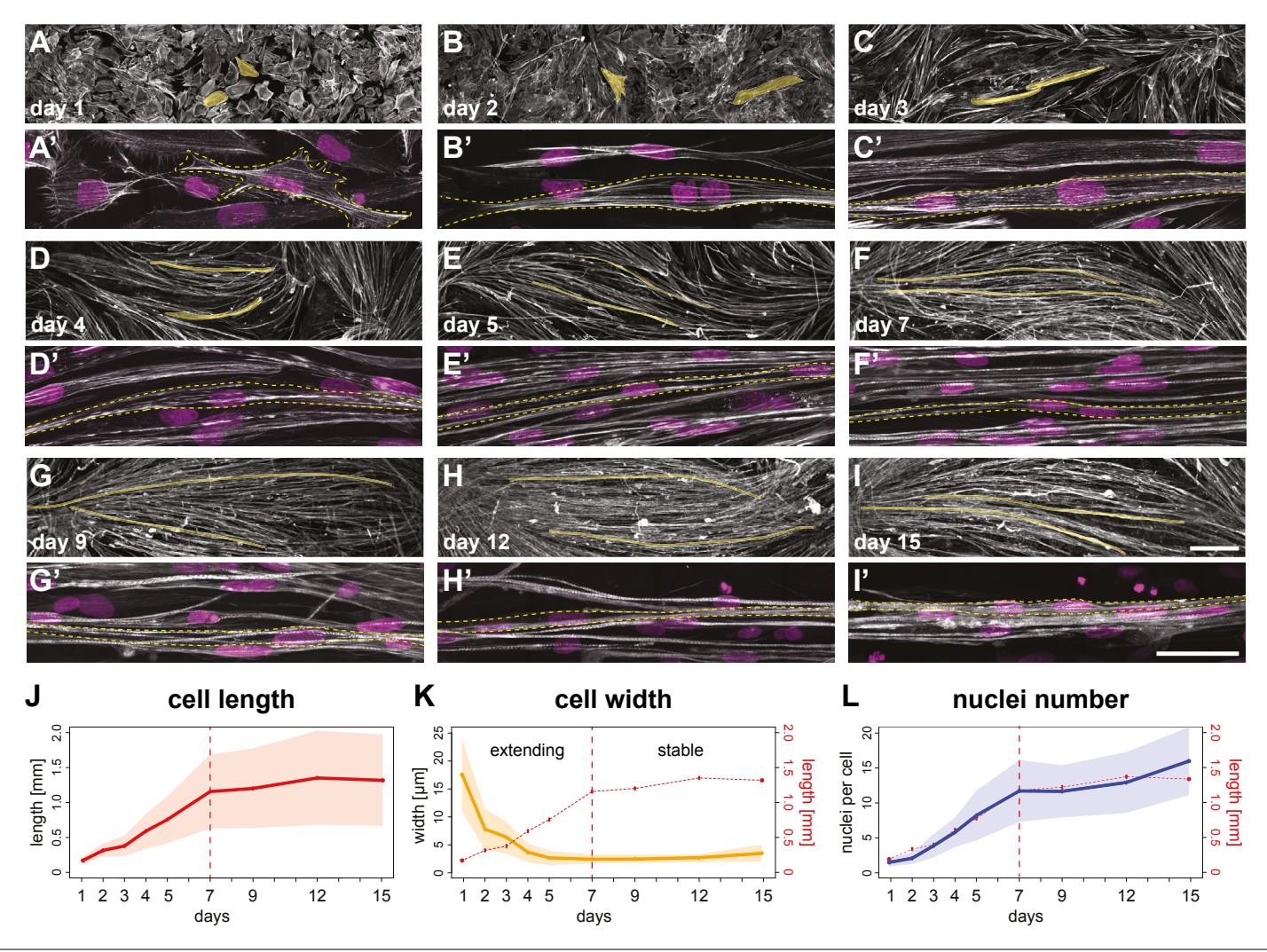

**Figure 1.** A time course of human induced pluripotent stem cell (iPSC)-derived myofiber differentiation in a minimalist two-dimensional (2D) culture. (**A–I**) ×10 views of phalloidin staining of human iPSC-derived myofibers. Yellow shades highlight two randomly selected myocytes, myotubes, or myofibers. (**A'-I'**) ×100 views of individual myocytes, myotubes, or myofibers. Rhodamine-phalloidin in grey, DAPI in magenta. Yellow dashed lines highlight individual myocytes/myotubes/myofibers. (**J**) Length of myocytes/myotubes/myofibers over time (n=35, 42, 4695, 3765, 4191, 2439, 2061, 1880, 1883 myofibers per reported day, respectively). Dots and solid line show the mean of each day, while shaded area shows the standard deviation. Vertical dashed line shows the transition point at day 7 from the elongation phase to the stable phase. (**K**) Width of myocytes/myotubes/myofibers over time (n=45, 50, 111, 191, 175, 297, 340, 315, 201 myofibers per reported day, respectively). (**L**) Number of nuclei per myofiber over time (n=35, 42, 76, 81, 85, 77, 62, 74, 66 myofibers per day). In (**K and L**), dots, shaded area, and solid line follow convention in **J**, with graph in **J** is reproduced in red. Scale bars: 250 µm in **A-I**, 50 µm in **A'–I'**. Source data available for **J–L**.

The online version of this article includes the following video, source data, and figure supplement(s) for figure 1:

**Source data 1.** Table containing source data from *Figure 1*.

**Figure supplement 1.** Dynamics of myocyte fusion and myotube elongation.

**Figure 1—video 1.** Phase-contrast movie of live myoculture showing a myocyte fusing into an already elongated myotube, and the elongation of a myotube.

https://elifesciences.org/articles/76649/figures#fig1video1

defined regions of disorder in the cultures with a nematic order below a threshold of 0.5. The fraction of disordered regions decreases over time (*Figure 2—figure supplement 1I*), further supporting an expansion of aligned regions.

In order to better understand the development of individual myofiber bundles in the entire cultures, we manually traced a total of 34,962 individual myotubes or myofibers and applied an automated

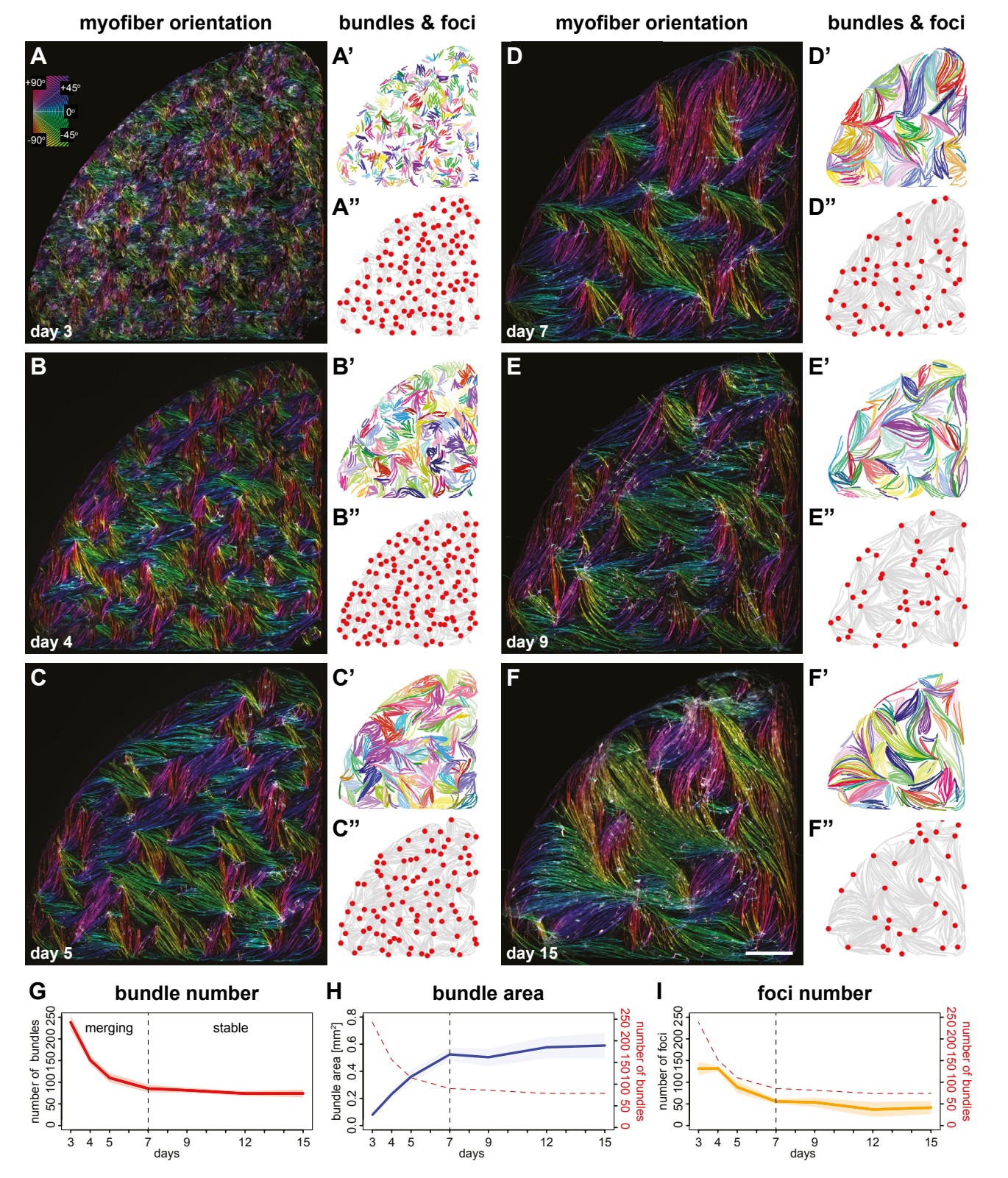

**Figure 2.** Myofibers self-organise into bundles that consolidate to stable sizes. (**A–F**) Rhodamine-phalloidin staining pseudo-coloured according to local fiber orientation by OrientationJ. Inset shows a colour wheel used by OrientationJ. (**A'–F'**) automatically segmented myofiber bundles from manually traced myofibers, colour-coded by bundle identity. (**A''–F''**) manually traced myofiber bundle end foci (red dots) superimposed on traced myofibers (grey lines reproduced from **A'–F'**). (**G**) Number of myofiber bundles over time (n=7 cultures per stage). Solid line shows the mean of each

*Figure 2 continued on next page*

*Figure 2 continued*

day, while shaded area shows the standard deviation. Vertical dashed line shows the transition point at day 7 from a merging to a stable phase. (**H**) Area of myofiber bundles over time (n=7 cultures per stage). (**I**) Number of myofiber bundle end foci over time (n=3 cultures per stage). Myofiber bundle number from G is overlayed in red. Scale bar: 1 mm. Source data available for **G–I**.

The online version of this article includes the following video, source data, and figure supplement(s) for figure 2:

**Source data 1.** Table containing source data from *Figure 2*.

**Figure supplement 1.** Vector field analysis of myofiber bundle alignment.

**Figure supplement 2.** Dynamics of myofiber bundles.

**Figure 2—video 1.** Phase-contrast movies of a live myoculture showing two myofiber bundles merging.

https://elifesciences.org/articles/76649/figures#fig2video1

Hausdorff distance clustering algorithm to objectively define distinct individual myofiber bundles (*Figure 2A*, see Materials and methods). This analysis revealed that the number of bundles decreases within 4 days from about 240 bundles (in each 0.35 cm$^2$ well) at day 3 to about 75 bundles at day 7 (*Figure 2A'–F' and G*). In turn, the average area occupied by each bundle increases about sixfold (*Figure 2H*). This strongly suggests that concomitant with myofiber elongation (see *Figure 1*), several smaller myofiber bundles merge into larger bundles.

Furthermore, myofiber ends present in the same bundle tend to point towards defined foci, which may serve as mechanical attachment sites to the surrounding matrix. In line with the reduction of bundle number over time, the number of these foci also decreases with similar temporal dynamics from about 140 foci at day 3 to about 50 foci at day 7 (*Figure 2A"–F"1*). This shows that these foci are dynamic and merge before day 7, suggesting that the myofiber bundles are not yet stably attached at their respective ends.

Next, we wanted to directly document the live dynamics of myofiber bundle and foci merging. Using a smaller culture format, we recorded the entire cultures by live phase-contrast imaging. Thus, we followed individual myofiber bundles over 4 days during differentiation and identified several examples in which one bundle elongates its extremity along a neighbouring bundle and finally merges with the neighbouring bundle to form shared attachment foci (*Figure 2—figure supplement 2*, *Figure 2—video 1*). This suggests that neighbouring myofiber bundles can offer orientation cues to nearby bundles during the dynamic bundle merging phase between day 3 to day 7 of the culture. This revealed that myofiber bundles can self-organise, potentially using mechanical orientation cues presented by their neighbours.

We noticed that myofiber bundles cease to merge after day 7 in our cultures (*Figure 2A–G*), coinciding with the end of myofiber extension (*Figure 1J*). Day 7 also demarcates a turning point for bundle area growth and foci fusion, both of which remain largely stable after day 7 (*Figure 2H1*). We conclude that myotubes and myofibers show a remarkable biphasic differentiation behaviour in vitro that is coordinated between cell and tissue scales. We hypothesised that the biphasic growth of myofibers and myofiber bundles may be caused by a switch in their mechanical status around day 7. This switch may further result in molecular changes in myofibers that enable myofibril and sarcomere formation.

## A transcriptional switch to enable myofibril and sarcomere assembly

As a first step to investigate the molecular changes at day 7 of differentiation, we analysed a previously generated transcriptomics dataset comparing similar proliferative cultures (using the identical SKGM-2 culture conditions) with days 7 and 15 of myogenic differentiation (using the identical KCTiP differentiation conditions) (*Al Tanoury et al., 2021*). After verifying the quality of the transcriptomics dataset with principal components analysis (*Figure 3—figure supplement 1*, *Supplementary file 1*), we performed Mfuzz clustering (*Kumar and E Futschik, 2007*), choosing 10 clusters as optimal for our dataset (see Materials and methods). Mfuzz clustering assigned all expressed genes to 1 of the 10 clusters which exhibit different expression dynamics (*Figure 3A*, *Supplementary file 1*, see Materials and methods). In search for clusters that may reflect the temporal dynamics of the biphasic myofiber extension behaviour, we found that three gene clusters are upregulated at day 7 of differentiation, while reaching a plateau or decreasing again at day 15 (clusters 1, 2, 7). Notably two of these clusters are enriched for muscle-specific GO-terms (*Figure 3A and B*). In particular, cluster 1 contains many

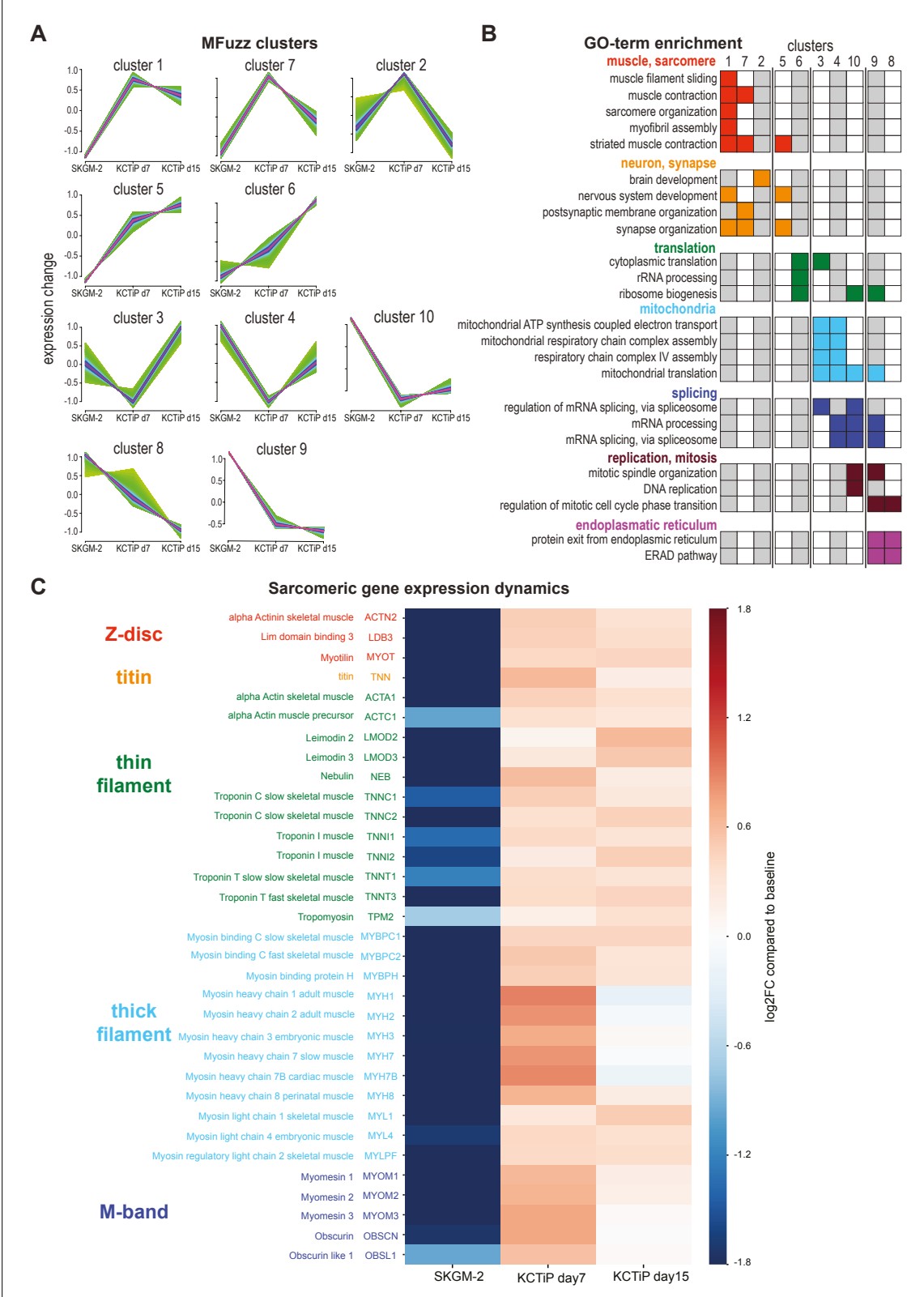

**Figure 3.** A transcriptional switch boosts sarcomeric gene expression. (**A**) Mfuzz clustering defined 10 distinct gene clusters with shared expression dynamics over the time course. (**B**) Selected enriched GO-terms for the respective clusters are displayed, a full list of GO-term enrichments is presented in *Supplementary file 1*. Note the two muscle-enriched Mfuzz clusters displaying a stark expression increase from undifferentiated to day 7. (**C**)

*Figure 3 continued on next page*

*Figure 3 continued*

Heatmap showing expression dynamics of various key sarcomeric genes comparing undifferentiated, day 7 and day 15 myogenic cultures to baseline. Note the stark expression increase of all components until day 7.

The online version of this article includes the following figure supplement(s) for figure 3:

**Figure supplement 1.** Quality control of the transcriptomics dataset.

genes coding for sarcomeric protein components, whose expression shows a strong upregulation at day 7 and then plateaus until day 15 (*Figure 3A and B*, *Supplementary file 1*). Furthermore, cluster 5, which is also strongly upregulated at day 7 and remains high at day 15, is also enriched for striated muscle contraction GO-terms (*Figure 3A and B*, *Supplementary file 1*). In contrast, components of the mitochondrial respiratory chain are only upregulated after day 7 and hence enriched in clusters 3 and 4 (*Figure 3A and B*, *Supplementary file 1*). This indicated that the sarcomere components are selectively upregulated at day 7, coinciding with the observed myofiber bundle consolidation.

Sarcomere assembly requires the concerted action of a large number of sarcomeric components (*Lange et al., 2006*). Closer inspection of the highly expressed and hence relevant genes coding for sarcomere components revealed that indeed all of them are strongly upregulated, many of them more than 10-fold at day 7 compared to proliferating cultures (*Figure 3C*, *Supplementary file 1*). This includes thin filament components actin A1, nebulin, leimodin 2, 3, and various troponin subunits, as well as thick filament components myosin heavy chain 1, 2, 3, 7, 8, myosin light chain 1, 3, and myosin binding protein C1, 2, and finally the titin protein, the Z-disc components α-actinin 2, the ZASP homolog LDB3, and the M-band core members obscurin and myomesin 1, 2, and 3 (*Figure 3C*). Many of these sarcomeric components are not further upregulated comparing day 7 to day 15 and hence are members of clusters 1 and 7, while expression of some does slightly increase further and hence these are members of cluster 5 (*Figure 3A and C*, *Supplementary file 1*). This suggests that the transcriptional switch inducing expression of the sarcomeric components at day 7 correlates well with the biphasic morphogenesis behaviour of myofibers reaching a stable length after day 7.

## Periodic sarcomeres emerge as myofibers approach stable lengths

The emergence of stable attachment foci during myotube differentiation and the strong induction of sarcomere gene expression at day 7 motivated a careful analysis of myofibrillogenesis and sarcomerogenesis in the differentiating cultures. We immunostained the cultures to visualise actin (with phalloidin), muscle myosin (with a myosin heavy chain antibody recognising all MyHC isoforms; *Webster et al., 1988*) and titin (with an N-terminal titin epitope antibody located at the sarcomeric Z-disc; *Mayans et al., 1998*) and manually segmented 810 myotubes or myofibers. Each fiber was then assigned to one of four morphologically distinct categories (*Figure 4A*): first, 'myotubes' display very low levels of titin in a salt-and-pepper pattern, muscle myosin levels are also low and actin does not yet form long continuous myofibrils. The lack of myofibrils is substantiated by the absence of periodic myosin or titin patterns on actin filaments (*Figure 4A and B*). Thus, myotubes do not yet contain myofibrils. Second, 'immature myofibers' display higher levels of titin and myosin, both of which are recruited onto continuous actin structures, which do not yet display an obvious periodicity (*Figure 4A and B*). Thus, immature myofibers contain immature myofibrils that are not yet periodic. Third, 'transitional myofibers' show high titin and myosin levels, both of which display a periodic pattern on long continuous periodic actin structures (*Figure 4A and B*). Hence, transitional myofibers contain long periodic myofibrils built from chains of immature sarcomeres. Fourth, 'mature myofibers' display strongly periodic patterns of titin, myosin, and actin across the entire width of the myofiber (*Figure 4A and B*). Thus, mature myofibrils in mature myofibers have either aligned or laterally expanded to form large Z-discs. Generally, the patterns appeared homogenous throughout most of the length of the myotube or myofiber; hence, we were able to categorise each of them into a single morphological category.

Having these four categories in hand, we classified all the 810 traced myotubes and myofibers across the entire time course to gain insights into the evolution of myofiber morphology at the population level (*Figure 4C*). Myotubes and immature myofibers are largely present before day 7 in the differentiating cultures, whereas transitional and mature myofibers dominate after day 7 (*Figure 4C*).

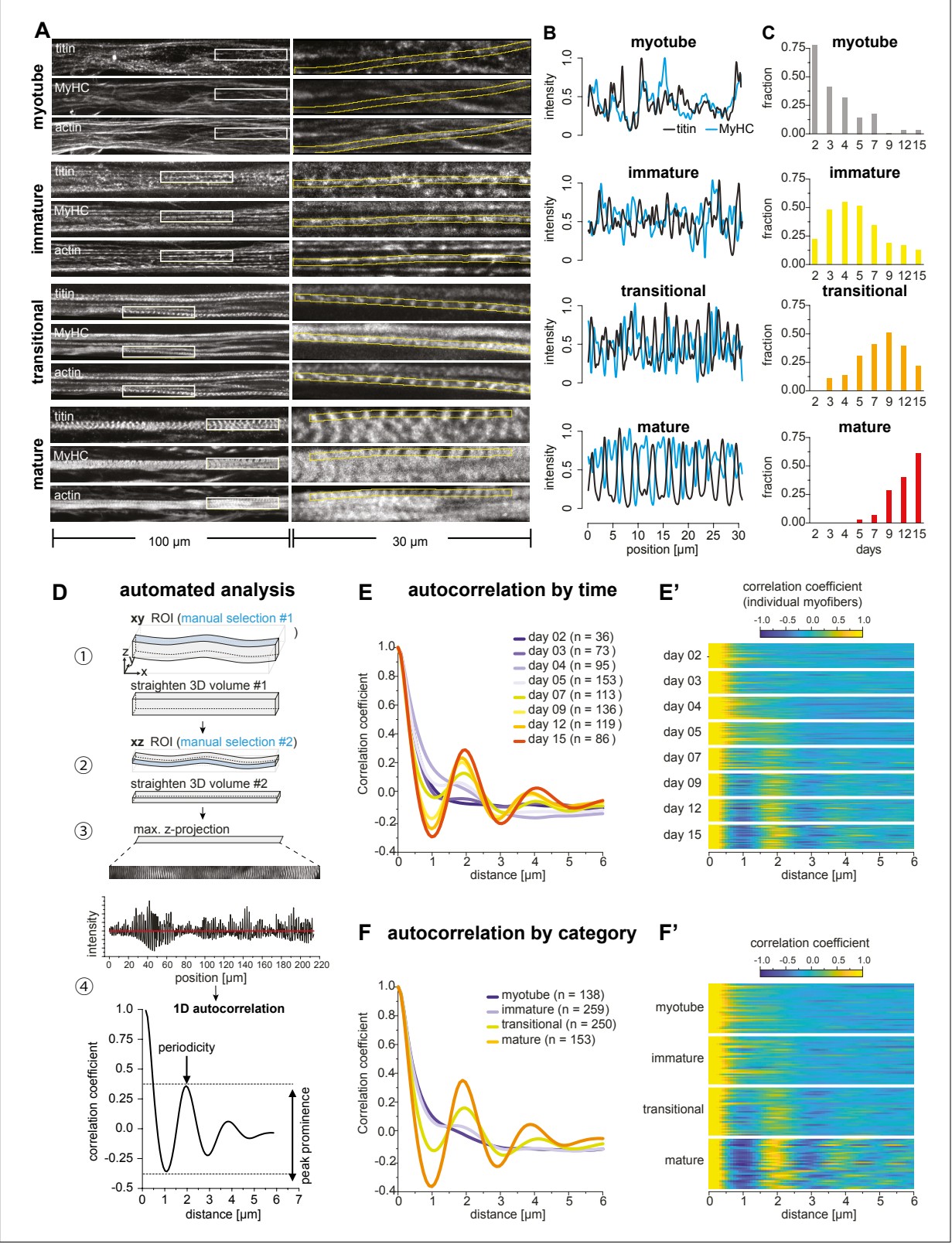

**Figure 4.** Sarcomeres emerge as myofibers approach stable lengths. (**A**) Multi-channel fluorescent images of four morphological categories (myotubes, immature, transitional, and mature myofibers). Left column shows a view of 100 μm wide, right column a zoomed view of 30 μm boxed area. Immunofluorescence against titin N-terminus, muscle myosin heavy chain (MyHC), and phalloidin staining of actin are shown. (**B**) Normalised one-dimensional (1D) fluorescent intensities for titin (black) and myosin (blue) from ROIs in right column in A. Note the emerging periodicity in transitional

*Figure 4 continued on next page*

*Figure 4 continued*

fibers. (**C**) Distribution of each morphological category during the time course. (**D**) Semi-automatic 1D autocorrelation analysis pipeline. Step ①: Straightening manual ROI in xy dimension. Step ②: Straightening ROI from Step ① in xz dimension. Step ③: Generating 1D intensity profile along myofiber. Step ④: Autocorrelation function (ACF) analysis with 1D intensity profile from Step ③. Presence of a secondary peak indicates the existence of periodic signal, with peak position indicating period length and peak prominence reflecting period regularity. (**E**) Average ACF for each time point. (**E′**) Randomly selected ACFs for each time point (n=36 each). (**F**) Average ACF for each morphological category. (**F′**) Randomly selected ACFs for each category (n=36 each). Source data available for **C**.

The online version of this article includes the following source data for figure 4:

**Source data 1.** Table containing source data from *Figure 4*.

This fits well with the observed biphasic behaviour described above and suggests that periodic sarcomeres largely assemble around day 7, the time when myofibers become stably attached.

In order to analyse the time course of sarcomere assembly in an unbiased manner, we developed a semi-automatic image analysis pipeline to quantitate periodic titin signal intensity profiles in a 3D volume extending along a length of up to 200 µm using an autocorrelation function (ACF) analysis (*Figure 4D*, see Materials and methods). After straightening the segmented myofibril to obtain a 1D titin intensity profile, we applied an ACF and monitored secondary ACF peaks position and prominence as a read out of the regularity of the pattern (*Figure 4D*; *Dehapiot, 2022*). We applied this pipeline to all myotubes and myofibers fixed at different time points in the differentiation time course. When averaging all ACFs for each time point, we found a secondary peak first emerging at days 5 and 7, with a period length of about 2 µm (*Figure 4E*). This sarcomere length fits well with the length of human foetal muscle sarcomeres (*Racca et al., 2013*). The peak prominence becomes higher at days 9, 12, and 15 (*Figure 4E*), demonstrating that the periodicity and thus the sarcomeres mature at later stages. This is also visible from individual ACFs in randomly selected myotubes or myofibers from each time point (*Figure 4E′*). Finally, we sorted the ACFs based on our manually assigned morphological categories and confirmed that immature myofibers do not yet display any obvious periodicity. Periodicity is however clearly detected in transitional and mature myofibers (*Figure 4F and F′*). We conclude that periodic sarcomeres assemble in transitional myofibers around days 5–7. This coincides with the time when myofibers have formed stable bundles with stable attachment foci (*Figures 1 and 2*). This correlation of morphological transitions across molecular, cellular, and tissue scales led us to hypothesise that mechanical signals may serve as a common denominator of morphological organisations.

## Myofiber bundling promotes sarcomere assembly

One important mechanical signal for sarcomerogenesis is tension (*Weitkunat et al., 2014*). We hypothesised that tension is modulated by myofiber bundling and hence bundling efficiency may directly impact sarcomere assembly. To test this hypothesis, we seeded myoblasts at varying densities from 2.5k to 20k cells per cm² and assayed bundling efficiency (all previous experiments were performed at a density of about 10k cells per cm²). We found that higher seeding density indeed accelerates myofiber bundling, as quantified by an increase in nematic length found at 10k and 20k seeding densities compared to lower densities at days 3 and 5 (*Figure 5A and B*, *Figure 5—figure supplement 1*). This is consistent with the hypothesis that fiber-fiber interactions are important for efficient myofiber bundling.

As we found that sarcomerogenesis correlates with the appearance of large stable myofiber bundles (at 10 k seeding density, see *Figure 2* and *Figure 4E*), we asked if changing the bundling speed will impact the dynamics of sarcomere assembly. By performing a similar ACF analysis of titin intensity as above we found that titin periodicity at day 5 is strongly increased in 10k and 20k cultures compared to lower density cultures (*Figure 5C–E*). At 20k density we found a few fibers that even show a mature titin pattern as early as day 3 (*Figure 5C*). These data support the hypothesis that myofiber bundling promotes sarcomere assembly, possibly by effective tension built-up in the larger myofiber bundles.

## Sarcomeres emerge simultaneously over broad regions in myofibers

In *Drosophila* muscles mechanical tension triggers the simultaneous assembly of periodic myofibrils across the entire length of the myofiber (*Weitkunat et al., 2014*). If mechanical tension also provides a long-range coordination signal in human myofiber cultures, then one would predict that sarcomeres also emerge across broad regions after stable attachment.

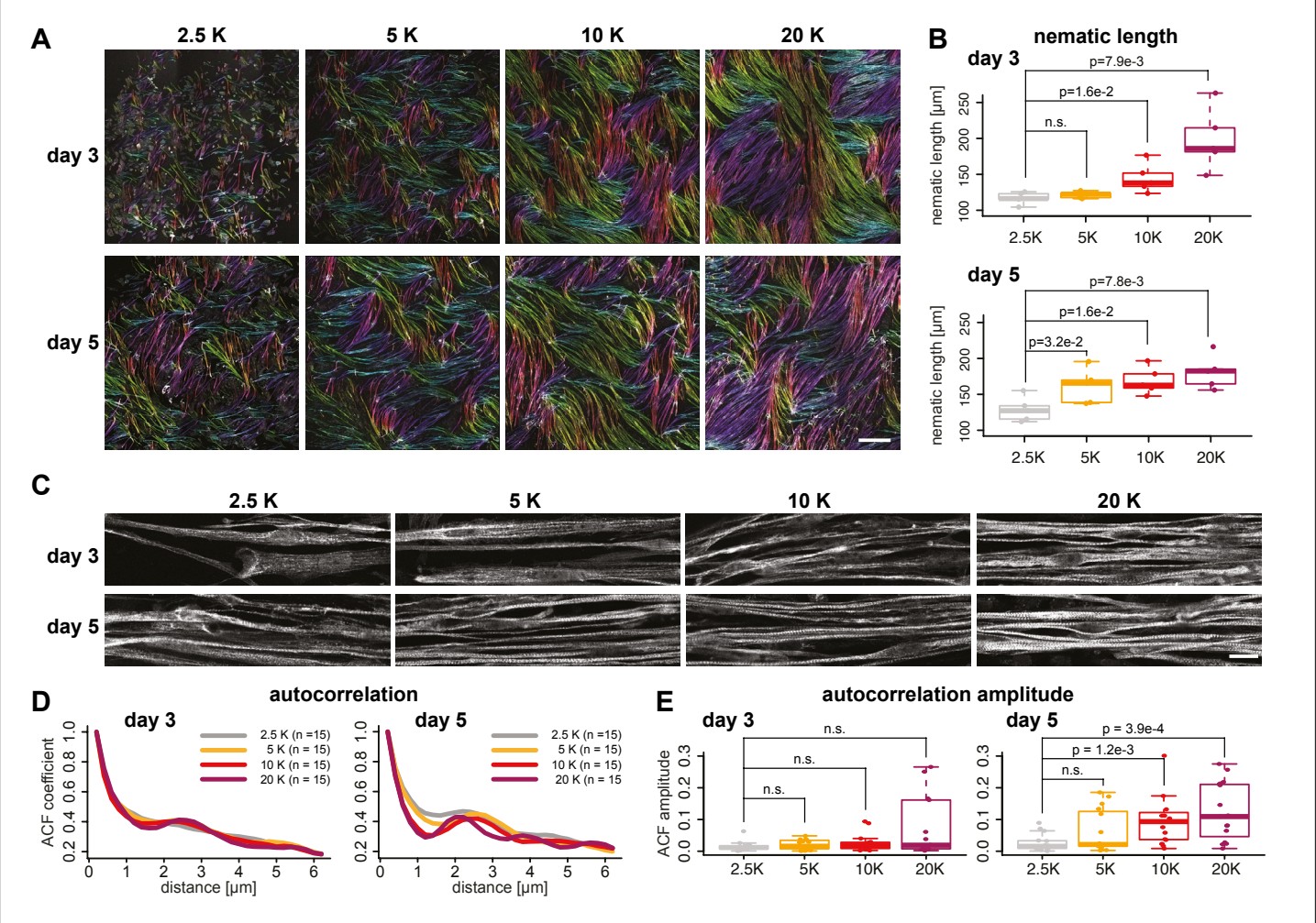

**Figure 5.** Myofiber bundling promotes sarcomerogenesis. (**A**) Rhodamine-phalloidin stainings of days 3 and 5 myofibers seeded at 2.5, 5, 10, and 20 k/ cm$^2$, pseudo-coloured according to local fiber orientation by OrientationJ (orientation colour-coding as in *Figure 2A–F*). (**B**) Nematic correlation length $\eta_L$ of conditions shown in **A**, n=5 each. (**C**) Immunofluorescence against titin N-terminus for conditions shown in **A**. (**D**) Average autocorrelation function (ACF) for conditions shown in **C**, n=15 each. (**E**) Amplitude of secondary ACF peaks for conditions shown in **D**, n=15 each. Scale bars: 500 µm in **A**, 20 µm in **C**. Multiple pair-wise t-test. Source data available for **B, E**.

The online version of this article includes the following source data and figure supplement(s) for figure 5:

**Source data 1.** Table containing source data from *Figure 5*.

**Figure supplement 1.** Vector field analysis of myofiber bundle alignment at different seeding densities.

In order to study how the first periodic myofibrils in human fibers are built, we performed local ACF analysis aiming to analyse sarcomere periodicity in 10-µm-wide windows sliding along the 'straightened' 1D titin intensity profiles ranging from 90 to 200 µm in length. This generated a 2D periodicity map for each myofiber, displaying the regularity of the pattern, measured by the peak prominence (*Figure 6A*, white line represents peak prominence), and distance between peaks (*Figure 6A*). To minimise artefacts generated by our straightening procedure, we optimised the peak prominence threshold and allowed for short gaps (below 6 µm in length; see Materials and methods). This enabled us to quantify the periodicity of titin distribution in all our 810 myotubes and myofibers (red lines in *Figure 6A*). As expected from the global ACF analysis shown in *Figure 4*, myotubes display almost no titin periodic region above the threshold, confirming that they do not contain any myofibrils or sarcomeres (*Figure 6A and B*). Similarly, immature myofibers contain on average less than one periodically patterned region per 100 µm, again verifying that they do not contain periodic myofibrils yet (*Figure 6A and B*). Therefore, myotubes and immature myofibers have not yet assembled periodic

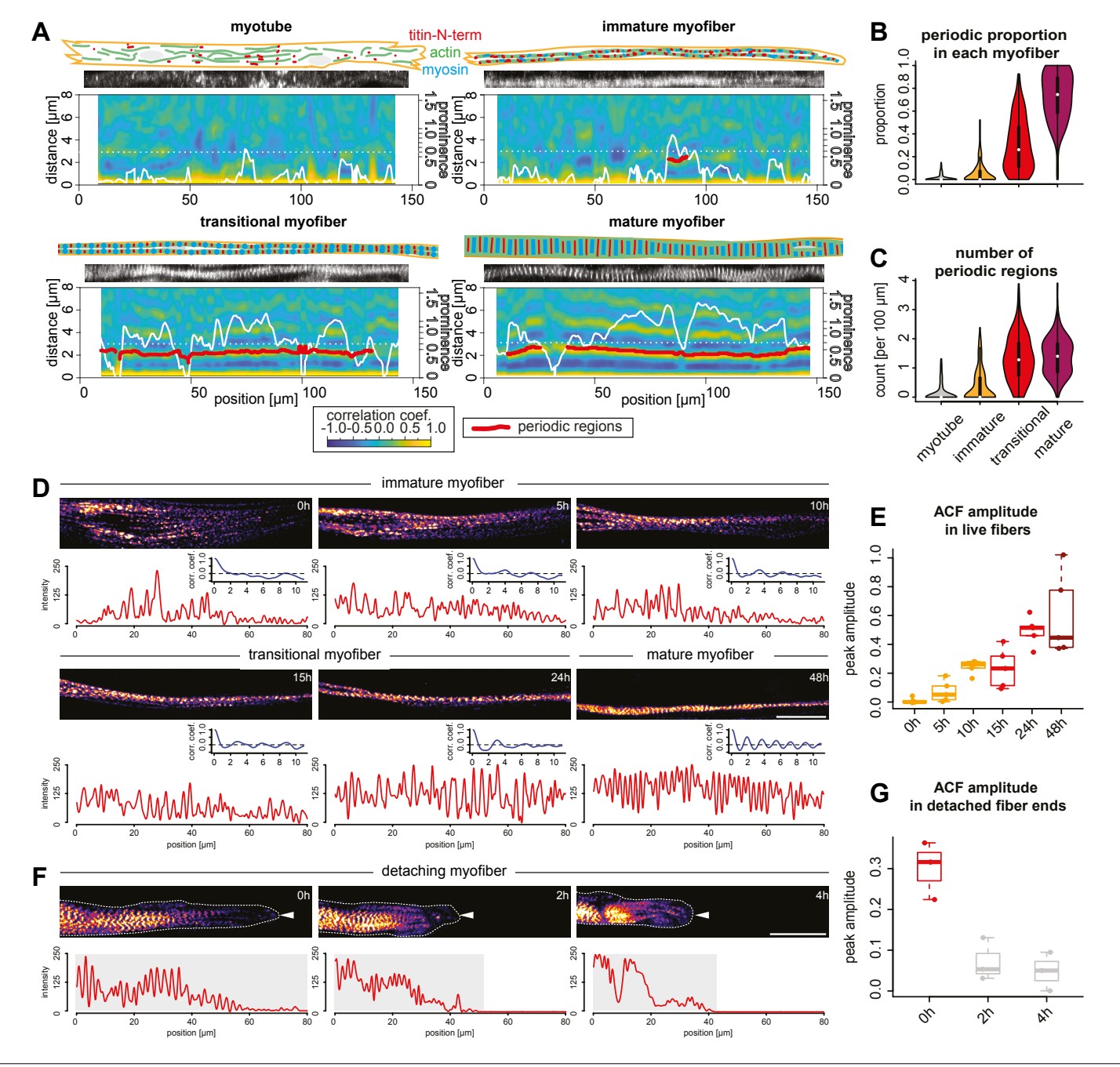

**Figure 6.** Sarcomeres emerge simultaneously over broad regions within myofibers. (**A**) Local autocorrelation function (ACF) analysis for each morphological category (myotubes, immature, transitional and mature myofibers). For each panel, top: schematic model of the myotube and myofiber stage with actin, myosin, and titin as indicated; middle: a straightened image of titin immunofluorescence; bottom: a two-dimensional (2D) colourmap of local ACF. x-Axis shows the position along the myofiber, y-axis shows the ACF distance. Superimposed white line shows prominence of the secondary peak along the myofiber. Superimposed red segments highlight secondary peak positions classified as periodic (see Materials and methods). (**B–C**) Violin plots of proportion of total length of detected periodic regions over whole lengths of myofiber (**B**) and number of detected periodic regions (**C**) for each myofiber from each category. Black bars mark the distance between first and third quadrant. White dots mark the median values. (**D**) Stills from a time-lapse movie showing a myofiber during myofibrillogenesis (labelled with mKate2-α-actinin2, *Figure 6—video 1*). Note that the end of the movie the myofibrils display a periodic pattern over the entire traced length, corresponding to the morphology of a mature myofiber. In each panel, the myofiber is traced to produce the intensity profile (red) and an ACF (blue) below that panel. The amplitude of the blue curve is then used to calculate values in **E**. (**E**) Amplitudes of ACFs at different time points in myofibers undergoing myofibrillogenesis (n=5). (**F**) Stills from a time-lapse movie showing one myofiber end retracting (labelled with mKate2-α-actinin2 and outlined with white dashed line and a white arrowhead, *Figure 6—video 2*). Note that

*Figure 6 continued on next page*

*Figure 6 continued*

the periodicity of the initially periodic myofiber collapses. In each panel, the myofiber is traced to produce the intensity profile (red) below that panel. The amplitude of the autocorrelation from the shaded region is then used to calculate values in **G**. (**G**) Amplitudes of ACFs at different time points in collapsing myofibers (n=3). Scale bars: 20 μm. . Source data available for **B, C, E, and G**.

The online version of this article includes the following video and figure supplement(s) for figure 6:

**Figure supplement 1.** Distribution of patterned regions in randomly selected myotubes, immature, transitional and mature myofibers.

**Figure 6—video 1.** A time-lapse movie showing a myofiber labelled with mKate2-α-actinin2 undergoing myofibrillogenesis.

https://elifesciences.org/articles/76649/figures#fig6video1

**Figure 6—video 2.** A time-lapse movie showing end retraction in a myofiber labelled with α- mKate2-α-actinin2.

https://elifesciences.org/articles/76649/figures#fig6video2

myofibrils and hence do not yet contain sarcomeres. In contrast, our ACF analysis revealed that mature myofibers are periodically patterned across 80% of the analysed regions, showing mostly one or two large regions (*Figure 6A–C*). This demonstrates that mature myofibers indeed contain highly regular periodic myofibrils that span most of the myofiber.

Together, this suggests that periodic myofibrils first assemble in transitional myofibers. This is supported by the finding that ACF analysis scores about 30% of their length as periodic (*Figure 6A and B*). Consistently, we find that transitional myofibers contain an average of 1.5 patterned regions per 100 μm (*Figure 6C*), some developed already into very long continuous regions (>50 μm, *Figure 6A*). This verified that transitional myofibers are the stage at which periodic sarcomeres assemble over long distances.

We next investigated the distribution of periodic regions along the length of transitional myofibers. If sarcomeres assemble simultaneously along the myofiber, our detection threshold should be reached at random positions, homogeneously distributed along the myofiber. Indeed, we find that patterned regions in transitional myofibers are homogeneously distributed throughout the entire fiber with no bias to the centre or to distal ends of the fibers (*Figure 6—figure supplement 1*). Taken together, this strongly suggests that sarcomeres emerge rather simultaneously over broad regions to form long periodic myofibrils in human myofibers.

To directly visualise sarcomerogenesis in human muscle cultures live, we tagged the endogenous α-actinin2 protein with mKate2 using CRISPR-Cas9 technology (see Materials and methods). Performing fluorescent time-lapse imaging, we were able to track individual myofibers in low-density cultures for up to 3 days, which enabled us to visualise myofibrillogenesis directly (*Figure 6D*). As observed in fixed cultures, we found that initially unpatterned immature myofibers develop periodic myofibrils over a broad region of the myofiber (*Figure 6D and E*, *Figure 6—video 1*). This likely coincides with a strong increase in mechanical tension as we frequently found one end of the myofiber detaching from the substrate. Strikingly, detaching from the substrate results in a local collapse of the periodic myofibrils (*Figure 6F and G*, *Figure 6—video 2*). This further supports our hypothesis that mechanical tension is the long-range signal that instructs and coordinates sarcomere assembly across the entire myofiber.

## Mechanical tension build-up precedes sarcomere assembly

In order to directly assess the role of tension in triggering the transition from a dynamic cytoskeleton in extending myotubes to periodic myofibrils in stable myofibers, we quantified mechanical tension using laser nano-surgery. We severed entire myotubes and myofibers expressing either endogenously tagged α-actinin2 (*Figure 7A*, myotube and immature myofiber) or titin (*Figure 7A*, transitional and mature myofibers). The recoil can be visualised by plotting the position of the severed ends as kymographs over time (*Figure 7A*, *Figure 7—video 1*). After cutting, severed myotubes, as well as immature, transitional, and mature myofibers all displayed a recoil response of their severed ends. We conclude that mechanical tension is present along the long axis of myotubes and myofibers of all morphological categories. These observations also indicate that mechanical tension precedes sarcomere assembly and persists during sarcomere maturation.

Additionally, we found that transitional and mature myofibers frequently show fast motions as consequences of active contractions (*Figure 7—figure supplement 1A, B* and *Figure 7—video 2*). Similar active contractions have been observed when severing immature myofibrils in *Drosophila*, as

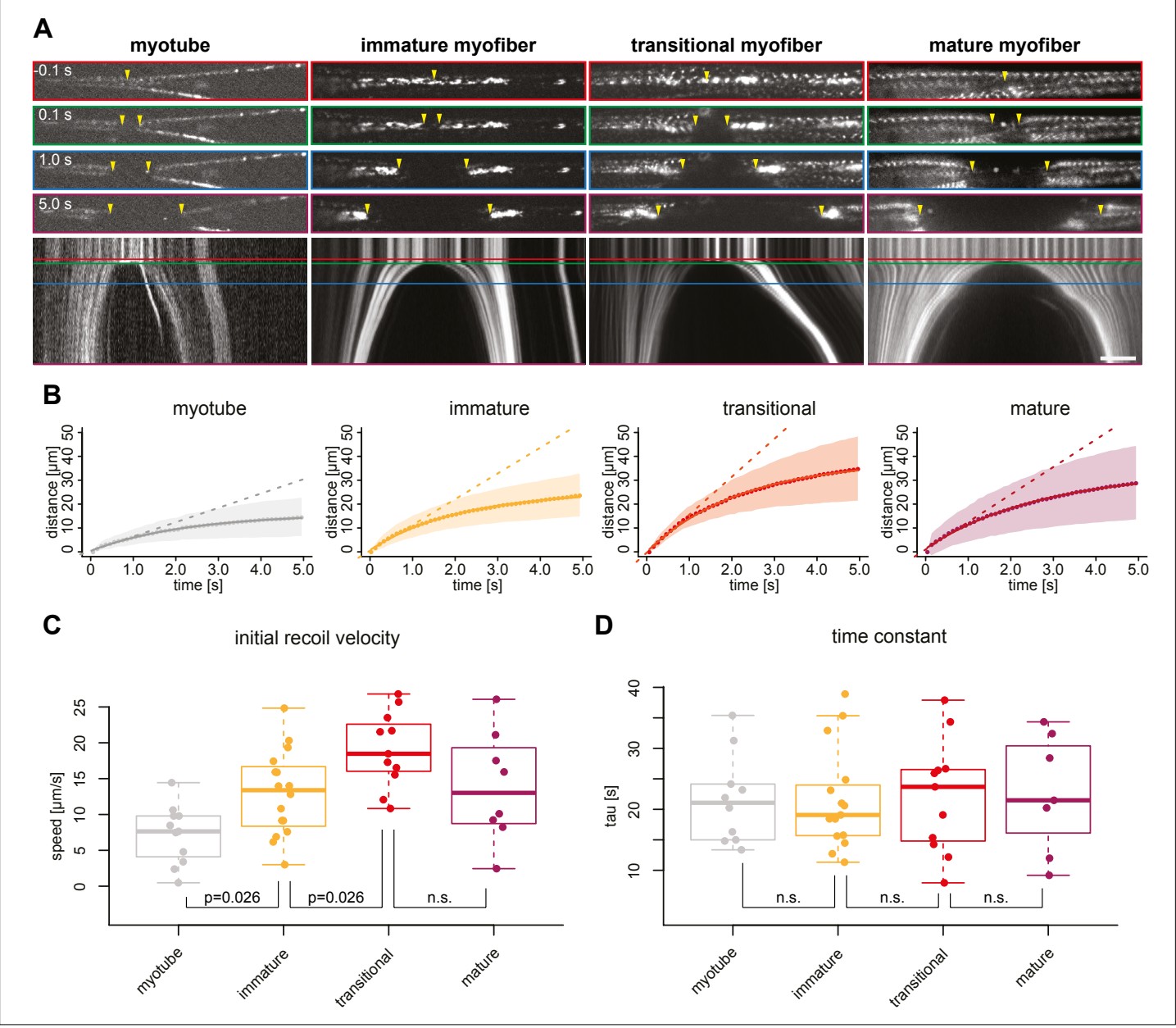

**Figure 7.** Mechanical tension precedes and persists during sarcomere formation. (**A**) Line laser ablation experiments for each morphological category (myotubes, immature, transitional and mature myofibers). Myotube and immature myofibers were labelled with mKate2-α-actinin2, while transitional and mature myofibers were labelled with GFP-titin. Top, upper middle, middle, and lower middle panels show 0.1 s prior to 0.1, 1.0, and 5 s post ablation, respectively. Bottom panel shows a kymograph of ablated myofibers. Coloured boxes correspond to coloured lines in the kymograph. Yellow arrowheads show sites of ablation and severed ends. (**B**) Quantification and mathematical modelling of recoil distances. Dots and shaded area show average and standard deviation of recoil distance, respectively. Dashed straight lines show a linear model fitted to estimate the initial recoil velocity. Solid curve shows viscoelastic model fitted to estimate the relaxation time $\tau$. (**C–D**) Box plot of initial recoil velocity (**C**) and relaxation time (**D**) from ablation experiments in each morphological category: n=11 (myotube), 16 (immature), 11 (transitional), and 8 (mature). Black bars mark the distance between first and third quadrant. White dots mark the median values. Multiple pair-wise t-test: n.s.: p>0.05. Scale bar: 10 μm. Source data available for **C** and **D**.

The online version of this article includes the following video, source data, and figure supplement(s) for figure 7:

**Source data 1.** Table containing source data from *Figure 7*.

**Figure supplement 1.** Contraction of myofibers revealed by point laser ablation.

**Figure supplement 2.** Tension on myofibrils revealed by point laser ablation.

*Figure 7 continued on next page*

*Figure 7 continued*

**Figure 7—video 1.** Line laser ablation experiments for myotube, immature, transitional, and mature myofibers.

https://elifesciences.org/articles/76649/figures#fig7video1

**Figure 7—video 2.** Contraction of myofibers during laser ablation.

https://elifesciences.org/articles/76649/figures#fig7video2

**Figure 7—video 3.** Recoil and collapse of myofibrils post point laser ablation.

https://elifesciences.org/articles/76649/figures#fig7video3

a consequence of laser-induced calcium release from the sarcoplasmic reticulum (*Weitkunat et al., 2017*) and have also been observed spontaneously in mature human myofibers after applying the same differentiation protocol (*Al Tanoury et al., 2021*). This demonstrates that transitional myofibers harbour already contractile sarcomeres.

If mechanical tension build-up is a trigger for periodic myofibril assembly, one would expect higher tension levels in transitional myofibers compared to earlier stages. As it is well established that the recoil velocity speed after the cut correlates with the amount of mechanical tension (*Behrndt et al., 2012*; *Hutson et al., 2003*), we tracked the recoiling ends of severed myotubes and myofibers (excluding the actively contracting ones) and fitted both a linear model and a viscoelastic model (*Figure 7B*) to calculate the initial recoil velocity (*Figure 7C*) and the viscoelastic relaxation time $\tau$ (*Figure 7D*), respectively. $\tau$ is comparable between all morphological categories, suggesting that they have comparable material properties. However, immature and transitional myofibers show a progressively higher initial velocity compared to myotubes (*Figure 7C*), demonstrating that they are under higher tension levels.

To probe tension directly within the assembling myofibrils, we performed nano-lesions to only severe individual myofibrils without affecting the rest of the cell, taking advantage of the high spatial precision of the pulsed infrared laser (see Materials and methods). We found that severed myofibrils are recoiling, demonstrating that they indeed are under tension (*Figure 7—figure supplement 2*, *Figure 7—video 3*). This strongly suggests that the assembling and maturing myofibrils are the major source of the mechanical tension present in the myofibers. We found that severing a mature myofibril leads to the collapse of its neighbouring sarcomeres (*Figure 7—figure supplement 2*, *Figure 7—video 3*), similar as in detaching myofibers observed in *Figure 6F*. In conclusion, these data demonstrate that attached myofibers build up mechanical tension prior to and during sarcomere formation, supporting our hypothesis that tension coordinates the assembly of long periodic myofibrils upon surpassing a threshold. These myofibrils need stable anchoring to prevent them from collapsing.

## Force-resistant attachment results in myofiber end clustering

In vivo, myofibers are stably attached to tendons cells that connect to the skeleton (*Lemke and Schnorrer, 2017*; *Schweitzer et al., 2010*). To mechanically anchor the myofibrils, the terminal Z-disc of each myofibril is connected to the cytoplasmic tails of integrins by a specialised actin-rich zone (*Green et al., 2018*; *Lemke et al., 2019*; *Sun et al., 2019*). Integrins in turn are localised to the cell membrane and their extracellular domains bind to various components of the muscle-tendon extracellular matrix (ECM) (*Brown, 2000*; *Hynes, 2002*). As our cultures do not contain prepatterned tendon-based attachments, we wondered how myofiber bundles develop stable attachments?

Integrins function as αβ-dimers. The human genome contains 18α- and 8β-integrin subunits, forming 24 functionally distinct dimers (*Ringer et al., 2017*; *Takada et al., 2007*). To identify which ones are likely relevant for attachment of muscle fibers in our cultures, we made use of the developmental gene expression time course and found that the fibronectin binding αvβ1-, α5β1-, and α6β1-integrins are the highest expressed integrins throughout the time course (*Supplementary file 1*). Additionally, α7-integrin is strongly upregulated at day 7, forming with β1 a laminin receptor (α7β1-integrin) (*Hynes, 2002*; *Plow et al., 2000*). These expression data fit with the severe muscle fiber and sarcomere assembly phenotypes of β1-integrin knock-out mice (*Brakebusch and Fässler, 2005*; *Schwander et al., 2003*) and the muscular dystrophy phenotype of α7-integrin knock-out mice (*Mayer et al., 1997*). Hence, we used β1-integrin as main adhesion marker together with paxillin, which is a force-sensitive component of adhesions (*Theodosiou et al., 2016*) and co-stained our cultures for titin and actin (*Figure 8A–D* and *Figure 8—figure supplement 1A–D*). Consistent with

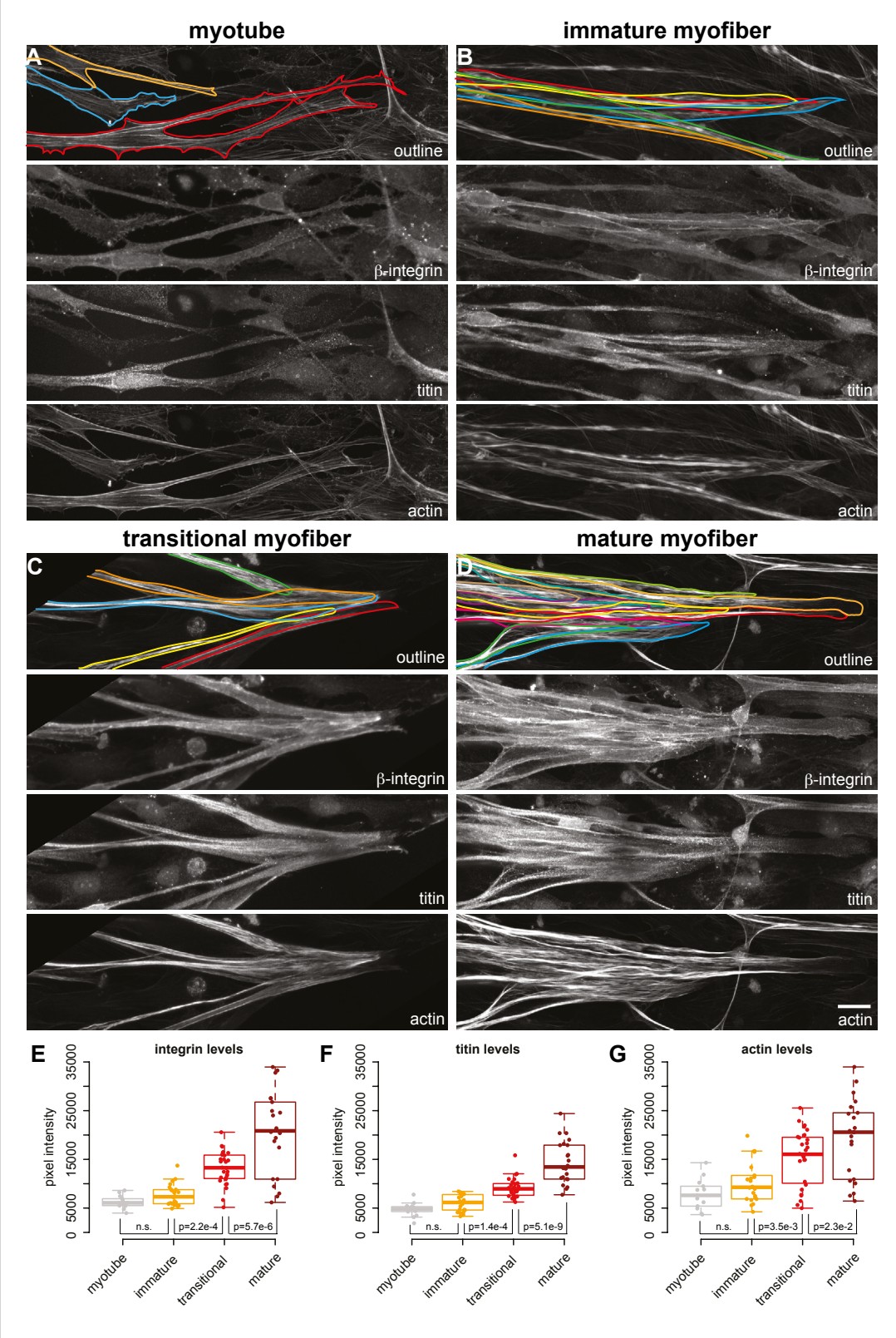

**Figure 8.** Attachment morphology during myofiber bundle maturation. (**A–D**) Immunostainings of myofibers with β1-integrin (upper middle), titin (lower middle) and actin (stained with phalloidin, bottom) in myotubes (**A**), immature (**B**), transitional (**C**), and mature myofibers (**D**). The top panel shows the outline of the myofibers superimposed on the actin channel. (**E–G**) Intensity of β1-integrin (**E**), titin (**F**), and actin (**G**) signals measured with a 50-pixel-wide line-ROI from myotubes (n=13), immature (n=22), transitional (n=28), and mature myofibers (n=22). Scale bars: 20 μm. Source data available.

*Figure 8 continued on next page*

*Figure 8 continued*

The online version of this article includes the following source data and figure supplement(s) for figure 8:

**Source data 1.** Table containing source data from *Figure 8*.

**Figure supplement 1.** Maturation of myofiber attachments revealed by additional attachment markers.

**Figure supplement 2.** Maturation of myofiber attachments revealed by additional attachment markers.

our transcriptomics data, we found that $\beta1$-integrin protein levels increase when the myofibers differentiate, coinciding with an increase in titin and actin levels (*Figure 8E–G*).

Closer inspection of high-resolution images showed that ends of myotubes contain an actin-rich network forming prominent filopodia with some integrin and paxillin-based focal adhesions (*Figure 8—figure supplement 1A*). This is consistent with the highly dynamic extension behaviour of myotube ends that we documented by live imaging (*Figure 1—figure supplement 1B*), resembling the dynamic focal adhesions of migratory cells (*Wehrle-Haller, 2012*). Similar integrin and paxillin foci are detected at the ends of immature myofibers, however integrin also localises along the entire outline of the immature myofiber (*Figure 8B*, *Figure 8—figure supplement 1B*). This suggests that the immature myofibers probe their environment over large areas, which is consistent with their above-described dynamics.

When the myofiber bundle foci stabilise and myofibers convert to transitional myofibers, we found a notable concentration of integrin and paxillin at the interface of neighbouring myofibers (*Figure 8C*, *Figure 8—figure supplement 1C*). This fiber-fiber localisation of integrin and paxillin further increases in mature myofibers, which frequently form large clustered foci (*Figure 8D*, *Figure 8—figure supplement 1D*). To better document the fiber-fiber proximity, we visualised the myofiber outlines with the cell surface marker NCAM1 and confirmed that immature as well as mature myofibers form immediate contacts at the bundle foci (*Figure 8—figure supplement 2*).

Taken together, our data suggest that the myofiber bundle foci of transitional and mature myofibers are indeed stabilised by force-resistant integrin-based fiber-fiber attachments to the surrounding ECM. These stable attachment foci can counteract the high tension produced in each transitional myofiber driving sarcomere assembly, and thus sarcomere assembly is coordinated with myofiber bundling at the supra-cellular level.

## Discussion
### Multi-scale self-organisation in human myofibers in vitro

In vivo, human muscle tissue displays a very ordered organisation ranging from the molecular scale of sarcomeric proteins to the tissue scale of myofiber bundles. This ensures that molecular forces produced by myosin motors translate efficiently into body movement. Here, we used a 2D culture system to investigate how muscle cells can organise themselves in the absence of any external cues. We discovered that myofibers show an autonomous multi-scale self-organisation when cultured in vitro (*Figure 9*).

At the tissue scale, myofibers self-organise into large bundles sharing one long axis, the future muscle contraction axis. Most myofibers within one bundle stably connect to one shared attachment focus on either end. This resembles the in vivo situation of muscle fiber bundles growing in the same direction and connecting to shared tendon and skeletal elements (*Schweitzer et al., 2010*; *Zelzer et al., 2014*).

At the cellular level, myotubes first elongate actively, then stabilise to a defined length, likely depending on the culture and matrix conditions (*Jensen et al., 2020*). At this stage, myofiber ends start to establish stable connections with their environment, coinciding with their transition from immature to transitional myofibers. Again, this behaviour recapitulates the in vivo situation, in which myotubes first elongate towards their future tendons, then they stably attach to them, before they organise their contractile myofibrils (*Gros et al., 2004*; *Kardon et al., 2003*; *Schnorrer and Dickson, 2004*; *Weitkunat et al., 2014*).

At the molecular level, sarcomeres emerge homogeneously with first periodic regions distributed across a long part of the myofiber after its stable attachment. These immature myofibrils then mature into chains of hundred or more periodic sarcomeres that span most of the mature myofiber. Further

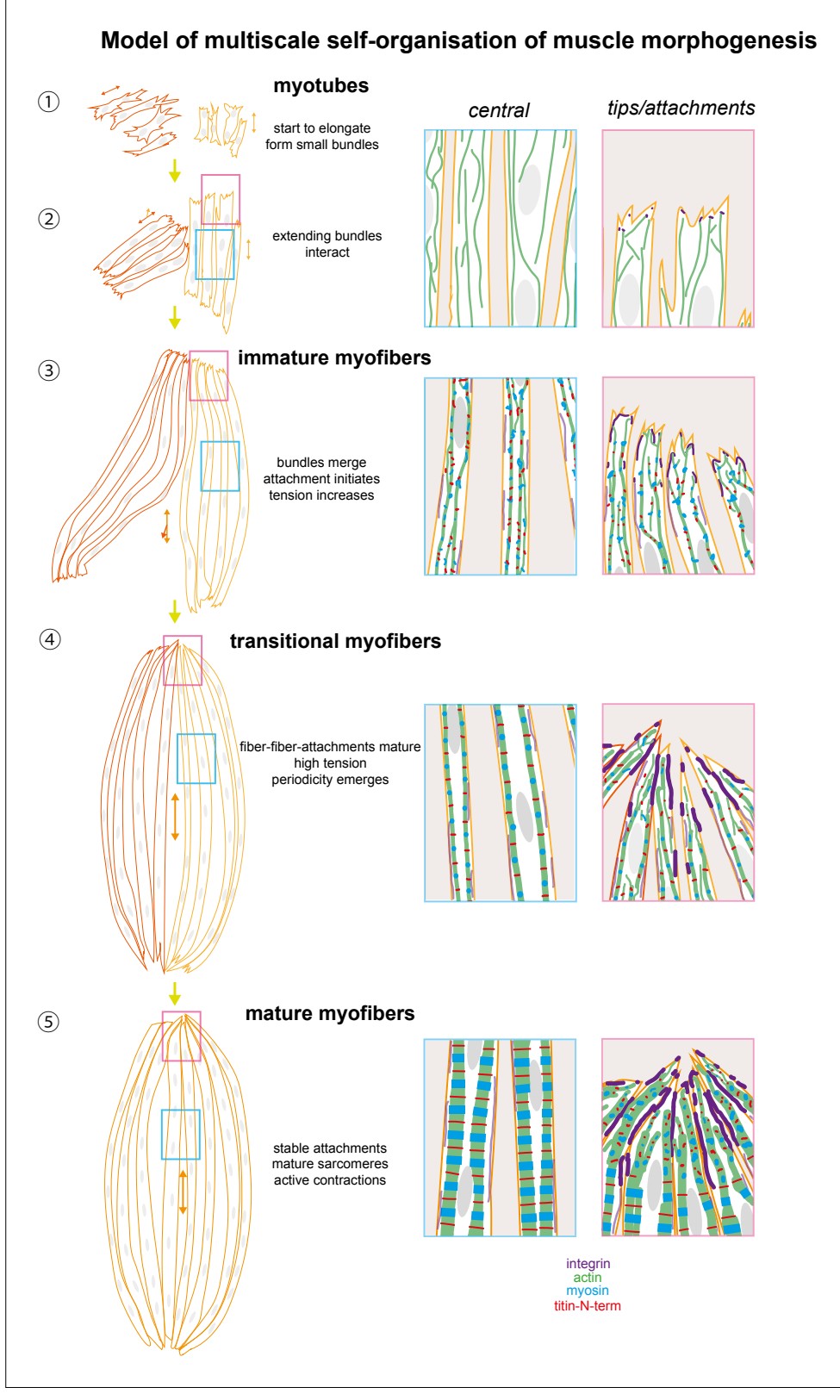

**Figure 9.** A tension-driven self-organisation model of human skeletal muscle morphogenesis across tissue, cellular, and molecular scales. See text for description.

maturation will result in the lateral expansion of the periodic myofibrils by lateral alignment of neighbouring myofibrils and incorporation of newly recruited components. A similar simultaneous myofibril assembly and maturation mechanism has been described in *Drosophila* muscles in vivo (*Weitkunat et al., 2014*; *Weitkunat et al., 2017*). Hence, our human muscle in vitro system does recapitulate various aspects of muscle morphogenesis in vivo.

The alignment of neighbouring myotubes to a shared contraction axis has also been found when cultured C2C12 mouse myoblasts were differentiated on patterned substrates with a defined axis (*Guillamat et al., 2020*; *Junkin et al., 2011*). The alignment of myofibers in this system depends on the compliance of the substrate, with very soft hydrogels resulting in highly parallel myotubes that do not converge to shared attachment foci (*Jensen et al., 2020*). Hence, self-organised alignment appears to be a general phenomenon of differentiating myotubes. However, prior to our work it was unknown how myotube alignment at the tissue scale coordinates with muscle and sarcomere morphogenesis at cellular and molecular scales in human muscle.

Here, we have presented evidence strongly supporting mechanical tension as one key signal to coordinate the self-organisation across scales. First, we detected different levels of mechanical tension during the time course of myofiber differentiation, starting with low tension in myotubes and resulting in high tension in the assembled transitional and mature myofibers. Second, we demonstrated that this tension is present across the assembling myofibrils. Since myofibrils span the entire muscle fiber, tension likely communicates rapidly across scales. Third, we found a strong correlation of tension increase with maturation of integrin attachments at the myofiber bundle foci. It is well established in many cell types that mechanical load on integrin adhesions exerted by the actin cytoskeleton results in their maturation to better withstand increased forces (*Austen et al., 2015*; *Bulgakova et al., 2017*; *Lemke et al., 2019*; *Sun et al., 2019*). Fourth, we demonstrated that more efficient myofiber bundling and thus likely faster tension build-up accelerates sarcomere assembly. Finally, we found that releasing tension either acutely by laser cutting a myofibril or by long-term detachment of a myofiber tip results in a collapse of the periodic myofibrils, which will likely result in a disassembly of the ordered structure. Altogether, this strongly suggests that mechanical tension is a key signal that coordinates morphogenesis in myofibrils, myofibers, and myofiber bundles.

## A self-organisation model across scales is coordinated by tension

How may tension coordinate muscle morphogenesis across scales? In our working model (*Figure 9*), myotubes elongate during the initial culture phase via protrusive ends. At this stage, they are already forming first mechanical contacts with neighbouring myotubes using filopodial extensions, which in response align to form a small and short myotube cluster sharing a common axis. The tension level is low and sarcomeric genes are not or little expressed (*Figure 9* – Steps ①, ②). As myotubes elongate further, they convert to immature myofibers: sarcomeric proteins begin to be expressed and actin filaments assemble into continuous myofibrils that are under higher tension but are still not periodic. The elongating myofiber bundles start to merge with neighbouring bundles to form a common axis and shared attachment foci (*Figure 9* – Step ③). The stable attachment enables a further increase in tension and the now high levels of sarcomeric proteins can assemble into periodic myofibrils throughout most of the myofiber (*Figure 9* – Step ④). The self-organisation process is likely myosin-driven, with myosin being responsible for the tension build-up and the ordering of the sarcomeric components, as has been shown in *Drosophila* (*Dasbiswas et al., 2018*; *Loison et al., 2018*) as well as in cardiomyocytes in vitro (*Chopra et al., 2018*; *Fenix et al., 2018*). Hence, we propose that tension-driven myofibril self-organisation is evolutionarily conserved across muscle types and tension can act as a compass to orient and coordinate between all the molecular components for coordinated assembly into continuous periodic myofibrils (*Lemke and Schnorrer, 2017*). We hypothesise that a critical tension threshold needs to be passed to trigger effective sarcomere assembly because individual myofibers progress at different speeds from myotubes to mature myofibers in our cultures.

Importantly, we found that integrin-based attachments are concentrated at the ends of the myofibers, in particular at the contact interfaces between neighbouring myofibers. Likely these integrins are mechanically engaged as is the case at muscle attachments in vivo (*Charvet et al., 2012*) thus allow the bundling of the myofibers to large bundles with one large attachment focus at each end. Such each of the foci couples mechanically to the ECM to resist the active myosin-based contractions that become prominent in mature myofibers (*Figure 9* – Step ⑤). Such active contractions are also

essential for myofibril maturation in *Drosophila* (*Weitkunat et al., 2017*), primary chick myofibers cultured in vitro (*Kagawa et al., 2006*), and in turn even for tendon and bone maturation in vivo (*Felsenthal and Zelzer, 2017*; *Shwartz et al., 2012*). Taken together, our model proposes a key role for mechanical tension to coordinate the self-organisation of muscle morphogenesis across scales.

## Tension sensing

Which molecules may sense tension to signal a coordinated assembly of myofibrils and myofibers into bundles? It is likely that tension is sensed at the myofiber level, as sarcomeres appear homogenously across the entire lengths of myofibers. A myosin-dependent self-organisation of ordered periodic actomyosin structures has even been observed in non-muscle cells (*Dasbiswas et al., 2018*; *Hu et al., 2017*). However, in contrast to non-muscle cells, the sarcomeric striations are stable over long time periods and have period length of 2–3 µm. This is achieved by titin, which 'locks' actin and myosin filaments in place, fulfilling its established ruler function (*Brynnel et al., 2018*; *Granzier et al., 2014*; *Linke, 2018*; *Tskhovrebova and Trinick, 2003*). In turn, titin is the major source of passive tension present in mature muscle and is responsible for the mechanical integrity of sarcomeres (*Li et al., 2020*; *Rivas-Pardo et al., 2020*; *Swist et al., 2020*). Hence, it is conceivable that the giant elastic titin molecules may serve as tension-sensitive catch bonds, whose incorporation into the myofibril subsequently triggers spatial segregation of actin and myosin domains into regular sarcomeres. However, molecular forces across titin during myofibril assembly have not yet been measured directly.

Another possible mechanism may exist at the ends of myofibers, where integrin-based attachment sites are mechanically linked to the actin cytoskeleton. Since myofiber elongation is mainly driven by actin dynamics of the myofiber ends (*Schnorrer and Dickson, 2004*; *Weitkunat et al., 2014*), a tension-sensing mechanism located at the myofiber ends could transmit the tension signal via the actin cytoskeleton to negatively regulate dynamics of myofiber ends and initiate stable attachment (*Richier et al., 2018*). Recent work from *Drosophila* muscle attachments revealed that high mechanical load on integrin does indeed recruit large amounts of the integrin adaptor talin, bridging from integrin to the actin cytoskeleton and the first sarcomeric Z-disc (*Bulgakova et al., 2017*; *Green et al., 2018*; *Luis and Schnorrer, 2021*; *Moser et al., 2009*). The large amounts of talin can then share the high mechanical load amongst all the talin molecules to achieve stable long-term attachment (*Lemke et al., 2019*). While we have not been able to detect costameric localisation of integrins in our cultured myofibers, likely due to their lack of maturity, we do not exclude a role of costameric integrins in mediating mechanical forces and promoting sarcomere formation (*Schwander et al., 2003*; *Sparrow and Schöck, 2009*), since it has been shown that costameres in mature myofibers play important roles in transmitting forces from the sarcomere to the sarcolemma and the ECM (*Peter et al., 2011*).

Finally, it is important that the transcriptional status of the differentiating myofiber fits with its morphological differentiation. Thus, tension may also feedback on the transcriptional machinery to boost expression of the sarcomeric components that are needed in large amounts to assemble myofibrils but are not present in extending myotubes. Such a feedback mechanism has recently been proposed in developing flight muscles, which use the mechanosensitive Hippo/YAP pathway to boost sarcomere gene expression before and during myofibril morphogenesis (*Kaya-Çopur et al., 2021*). Tension could also be sensed directly at the nuclear envelope, which is linked by the LINC complex to the actin cytoskeleton, inducing changes of nuclear shape and position during myofiber differentiation (*Janota et al., 2020*; *Wang et al., 2018*). Hence, it is conceivable that in the developing human muscles tension is sensed to boost sarcomeric gene expression. Such a mechanism would ensure that sarcomeres only assemble from their components when the myofibers are properly attached.

## Making even better muscle in vitro – towards a muscle-skeleton 'organoid'?

The 2D myofibers generated in vitro are still immature compared to large adult human muscle fiber bundles attached to tendons and bone in vivo. Our myofibers show highest expression of embryonic (MYH3) and neonatal (MYH8) myosin isoforms (see *Supplementary file 1*; *Schiaffino et al., 2015*; *Al Tanoury et al., 2021*) and their myofibrils remain rather thin. However, longer-term culture often results in myofiber detachment followed by degeneration, likely because the attachment sites are not stable enough to withstand the even higher forces produced by more mature fibers. This could be

partially solved by overlaying with second layer of Matrigel to provide additional mechanical support in 3D (*Roman et al., 2018*). The successful differentiation of contractile human neuromuscular organoids from neuromesodermal progenitors in 3D (*Faustino Martins et al., 2020*) also suggests that 3D support of the myogenic cultures is advantageous.

In summary, our current study has uncovered a remarkable transition from an initial population of dynamic, elongating myotubes to stable and contractile bundles of myofibers that is likely coordinated across tissue, cellular, and molecular scales via mechanical tension. Given that muscle-generated mechanical forces are crucial for optimal patterning and morphogenesis in bones and tendons (*Felsenthal and Zelzer, 2017*), it is tempting to hypothesise that mechanical tension may also coordinate the morphogenesis across muscle and connective tissue lineages, resulting in a self-organised musculoskeletal system. Introducing progenitors of tendon or other connective tissues at defined positions of the cultures will be a next step towards a minimal sufficient system that can recapitulate mature muscle-tendon-skeleton development in vitro.

# Materials and methods
## Materials availability statement
Any newly created material in this manuscript can either be accessed online or requested from the authors.

## Key resources table

| Reagent type (species) or resource | Designation | Source or reference | Identifiers | Additional information |
| --- | --- | --- | --- | --- |
| Gene | ACTN2 (human) | https://www.ncbi.nlm.nih.gov/gene/88 | HGNC:HGNC:164 | |
| Cell line | NCRM-1 (*Homo sapiens*) | RUCDR, Rutgers University; *Chal et al., 2016*; *Al Tanoury et al., 2021* | | |
| Cell line | GFP-titin (*Homo sapiens*) | Derived from PGP-1; *Sharma et al., 2018* | | |
| Cell line | mKate2-α-actinin2 | Derived from NCRM-1, this study | | |
| Antibody | Rabbit-anti-titin (N-term) | Sigma, Cat. HPA007042 | | 1 in 50 |
| Antibody | Mouse anti-MyHC | DSHB, Clone A4.1025 | | 5 ng/µL |
| Antibody | Rat anti-β1integrin | DSHB, Clone AIIB2 | | 5 ng/µL |
| Antibody | Mouse anti-paxillin | Invitrogen, Cat. AHO0492 | | 1 in 5 |
| Antibody | Anti-NCAM1 | DSHB, Clone 5-1H11 | | 5 ng/µL |
| Antibody | Goat anti-mouse Alexa405 | Invitrogen, Cat. A31553 | | 1 in 250 |
| Antibody | Goat anti-rabbit Alexa405 | Invitrogen, Cat. A31556 | | 1 in 250 |
| Antibody | Goat anti-rabbit Alexa488 | Life Technologies, Cat. A11034 | | 1 in 250 |
| Antibody | Goat anti-rat Alexa488 | Invitrogen, Cat. A11006 | | 1 in 250 |
| Antibody | Goat anti-mouse Alexa488 | Invitrogen, Cat. A11001 | | 1 in 250 |
| Antibody | Goat anti-rat Alexa633 | Life Technologies, Cat. A21094 | | 1 in 250 |
| Antibody | Goat anti-rabbit Alexa633 | Invitrogen, Cat. A21070 | | 1 in 250 |
| Sequence-based reagent | mKate2 dsDNA | Synthesised and obtained from Integrated DNA Technologies | | |
| Commercial assay or kit | Culture-insert 4 well in µ-dish 35 mm | ibidi, Cat. 80466 | | |
| Commercial assay or kit | Micro-insert 4 well in µ-dish 35 mm | ibidi, Cat. 80406 | | |
| Chemical compound, drug | SKGM-2 medium | Lonza, Cat. CC-3245 | | |
| Chemical compound, drug | ROCK inhibitor Y27632 | Tocris, Cat. 1254 | | 0.1% |
| Chemical compound, drug | Matrigel | Corning, Cat. 354277 | | 1 in 40 |
| Chemical compound, drug | DMEM/F12 medium | Thermo Fisher Scientific, Cat. 11320033 | | |
| Chemical compound, drug | Knock-out Serum Replacement | Invitrogen, Cat. 10828028 | | 2% |

*Continued on next page*

*Continued*

| Reagent type (species) or resource | Designation | Source or reference | Identifiers | Additional information |
|---|---|---|---|---|
| Chemical compound, drug | ITS | Life Technologies, 41400045 | | 1% |
| Chemical compound, drug | Pen/Strep | Life Technologies, Cat. 15140122 | | 0.2% |
| Chemical compound, drug | Chiron | Tocris Bioscience, Cat. 4423 | | 1 mM |
| Chemical compound, drug | TGF-β inhibitor SB431542 | Tocris, Cat. 1614 | | 10 mM |
| Chemical compound, drug | Prednisolone | Sigma-Aldrich, Cat. P6004 | | 10 mM |
| Chemical compound, drug | Gibco Fetal Bovine Serum | Thermo Fisher Scientific, Cat. A4766801 | | |
| Chemical compound, drug | Rhodamine-phalloidin | Invitrogen, Cat. R415 | | 1 in 500 |
| Chemical compound, drug | Alexa568-phalloidin | Invitrogen, Cat. A12380 | | 1 in 500 |
| Chemical compound, drug | Prolong Gold Antifade without DAPI | Invitrogen, Cat. P36934 | | |
| Chemical compound, drug | Prolong Gold Antifade with DAPI | Invitrogen, Cat. P36935 | | |
| Software, algorithm | CRISPOR tool | http://crispor.tefor.net | | |
| Software, algorithm | CENTURI_Mao_FDomMuscle | https://github.com/BDehapiot/CENTURI_Mao_FDomMuscle; (*Dehapiot, 2022*) | | |

## Human iPSC-derived 2D skeletal myoculture

The following human iPSC lines were used to perform myogenic differentiation: untagged NCRM-1 (RUCDR, Rutgers University), GFP-titin (derived from PGP-1, *Sharma et al., 2018*) and mKate2-α-actinin2 (derived from NCRM-1, this study, see below). The cell lines were tested to not contain any mycoplasma contamination.

Maintenance of human iPSCs and their primary myogenic differentiation were performed as described in *Chal et al., 2016*; *Al Tanoury et al., 2021*. For secondary differentiation, primary myogenic cultures derived from iPSCs were dissociated at 3 weeks post differentiation, filtered through a 70 and 40 µm cell strainer, and cryopreserved in custom cryomedia (*Chal et al., 2016*). After thawing in 37°C water bath, myogenic progenitors were resuspended in skeletal muscle growth medium (SKGM-2, Lonza, Cat. CC-3245) with 10 µM ROCK inhibitor (Y-27632 dihydrochloride, Tocris, Cat. 1254), and seeded at a density of 10 k/cm$^2$ (or at varying densities from 2.5 to 20 k/cm$^2$ in *Figure 5*) onto polymer coverslip-bottomed imaging dishes (culture-insert 4 well in µ-dish 35 mm, ibidi, Cat. 80466 for fixed experiments or micro-insert 4 well in µ-dish 35 mm, ibidi, cat. 80406 for live experiments) coated with Matrigel (Corning, Cat. 354277). After 24 hr, culture medium was replaced by SKGM-2 medium without ROCK inhibitor. Myogenic progenitors were allowed to proliferate for 1–2 days until reaching ~80% confluence, then induced for myogenic differentiation by replacing SKGM-2 medium with the KCTiP differentiation medium (*Al Tanoury et al., 2021*): DMEM/F12 medium (Thermo Fisher Scientific, Cat. 11320033) supplemented with 2% knock-out serum replacement (Invitrogen, Cat. 10828028), 1% ITS (Life Technologies, 41400045), 0.2% Pen/Strep (Life Technologies, Cat. 15140122), 1 µM Chiron (Tocris Bioscience, Cat. 4423), 10 µM TGF-β inhibitor SB431542 (Tocris, Cat. 1614), and 10 µM of prednisolone (Sigma-Aldrich, Cat. P6004). KCTiP differentiation medium was subsequently changed on days 1 and 2, and then was refreshed every other day until the end of the cell culture.

## Generation of the mKate2-α-actinin2 knock-in reporter line

Two different sgRNA targets were designed using the CRISPOR tool (http://crispor.tefor.net) surrounding the start site of human ACTN2 gene for N-terminal tagging (primers sequences are listed below). sgRNAs oligonucleotides were annealed and cloned into pSpCas9(BB)-2A-GFP. Donor plasmid was designed by NEBuilder (https://nebuilderv1.neb.com) using Gibson assembly in pUC10 backbone. 0.8–1 kb nucleotide sequence before the sgRNA cutting site as 5′-homology arm (HA), mKate2 and 0.8–1 kb nucleotide sequence after the sgRNA cutting site as 3′-homology arm were used to generate primers for cloning (see below). A silent mutation was introduced on each of the donor template (5′-HA and 3′-HA) in the PAM site to prevent the CRISPR from targeting the recombined sequences. 5′-HA and 3′-HA were PCR amplified using human genomic DNA. mKate2 dsDNA

was synthesised and obtained from Integrated DNA Technologies. Gibson cloning method was used to ligate each part to construct the full donor plasmid (New England Biolabs). Electroporation of the pSpCas9(BB)-2A-GFP and donor plasmids was performed with an Amaxa Nucleofector IIb device (Lonza). GFP-positive cells were isolated by fluorescence-activated cell sorting, clonally expanded, and then selected by PCR and validated through Sanger sequencing.

## Primer sequences:

sgRNA1:
Forward Primer: 5'CACCGCACGTAGTTGTACTGCACGC3'
Reverse Primer: 5'AAACGCGTGCAGTACAACTACGTGC3'
sgRNA2:
sgRNA target site: ACTGCACGCCGGGCTCTATC
Forward Primer: 5'CACCGACTGCACGCCGGGCTCTATC3'
Reverse Primer: 5'AAACGATAGAGCCCGGCGTGCAGTC3'
*5'HA (R)* (anneals pUC19): GACGAAAGGGCCTCGTGATAC
pUC19 (F) (anneals 5'HA): TATCACGAGGCCCTTTCGTCGAGACACGAGAGAGAG
mKate2 (R) (anneals 5'HA): CGCTCACCATCATGTACTCATCCTCGTC
5'HA (F) (anneals mKate2): TGAGTACATGATGGTGAGCGAGCTGATTAAG
3'HA (R) (anneals mKate2): ACTCCTCCTCCTGGATTCATCTGTGCCCCAGTTTG
mKate2 (F) (anneals 3'HA): GCACAGATGAATCCAGGAGGAGGAGTGG
pUC19 (R) (anneals 3'HA): GTCATCACCGAAACGCGCGAAAAAGTTCAGAGCGTTTGG
3'HA (F) (anneals pUC19): TCGCGCGTTTCGGTGATG

## Immunostaining of fixed myocultures

Myocultures in imaging dishes were rinsed one time in ×1 PBS before fixed in 4% PFA for 20 min at room temperature. Fixed myocultures were rinsed one more time in ×1 PBS, permeabilised in ×1 PBS supplemented by 0.1% Triton X-100 (Sigma-Aldrich, Cat. T8787) for 15 min at room temperature, then incubated in blocking buffer composed of ×1 PBS supplemented with 10% Gibco Fetal Bovine Serum (Thermo Fisher Scientific, Cat. A4766801) and 0.1% Triton X-100. Myocultures were subsequently incubated with appropriate primary antibodies diluted in blocking buffer at appropriate concentrations overnight at 4°C. Myocultures were then washed three times with ×1 PBS supplemented by 0.1% Tween 20, and incubated with appropriate fluorescently labelled secondary antibodies and phalloidin diluted ×1 PBS supplemented by 0.1% Tween 20 for 2 hr at room temperature. Myocultures were then washed three times with ×1 PBS supplemented by 0.1% Tween 20 (Sigma-Aldrich, Cat. P9416). The culture inserts were removed to allow mounting with a top coverslip and appropriate mounting media. Samples were sealed with nail polish and stored at 4°C until imaging.

The following primary antibodies were used: rabbit-anti-titin (N-term) (Sigma, Cat. HPA007042, 1:50), mouse anti-MyHC (DSHB, Clone A4.1025, 5 ng/µL), rat anti-β1integrin (DSHB, Clone AIIB2, 5 ng/µL), mouse anti-paxillin (Invitrogen, Cat. AHO0492, 1:5), mouse-anti-NCAM1 (DSHB, Clone 5-1H11, 5 ng/µL). The following secondary antibodies were used: goat anti-mouse Alexa405 (Invitrogen, Cat. A31553, 1:250), goat anti-rabbit Alexa405 (Invitrogen, Cat. A31556, 1:250), goat anti-rabbit Alexa488 (Life Technologies, Cat. A11034, 1:250), goat anti-rat Alexa488 (Invitrogen, Cat. A11006, 1:250), goat anti-mouse Alexa488 (Invitrogen, Cat. A 11001, 1:250), goat anti-mouse Alexa633 (Life Technologies, Cat. A21052, 1:250), goat anti-rat Alexa633 (Life Technologies, Cat. A21094, 1:250), goat anti-rabbit Alexa633 (Invitrogen, Cat. A 21070, 1:250). The following staining reagents were used: rhodamine-phalloidin (Invitrogen, Cat. R415, 1:500), Alexa568-phalloidin (Invitrogen, Cat. A12380, 1:500). The following mounting media were used: Prolong Gold Antifade without DAPI (Invitrogen, Cat. P36934), Prolong Gold Antifade with DAPI (Invitrogen, Cat. P36935).

## Fluorescence imaging of fixed myocultures

Myocultures were differentiated in polymer coverslip-bottomed imaging dishes (culture-insert 4 well in µ-dish 35 mm, ibidi, cat. 80466) coated with Matrigel and fixed as described above. Fluorescent images of fixed myocultures were acquired on a Zeiss LSM880-I-NLO inverted confocal microscope using the following objectives: EC Plan-Neofluar ×10/0.3, Plan-C Apochrome ×40/1.2 W, Apochromat ×100/1.4 Oil DIC. For images acquired at ×10 magnification, 6×6 tiles at zoom 0.6 were used to cover

the entire culture well, while single z-slices were acquired. For images acquired at ×40 or ×100 magnification, various tiles at zoom 1.0 were used to cover a larger area, with a z-step size of 0.3–0.5 µm.

## Phase-contrast time-lapse imaging of live myocultures

Myocultures were differentiated in polymer coverslip-bottomed imaging dishes (micro-insert 4 well in µ-dish 35 mm, ibidi, Cat. 80406) coated with Matrigel as described above. Phase-contrast time-lapse images of live myocultures were acquired on a Zeiss Axio Observer.Z1 inverted wide-field microscope with 5% $CO_2$ at 37°C using an EC Plan-Neofluar ×20/0.5 objective. 4×4 tiles at zoom 1.0 were used to cover a larger area. Images were acquired every 30 min, for up to 6 days. Closed culture dishes filled with ample media kept in a moisturised imaging chamber allowed for stable long-term imaging without perturbing the sample by media change. Software autofocus was used to keep the imaging plane constant.

## Fluorescence time-lapse imaging of myofibrillogenesis

Myocultures were differentiated in polymer coverslip-bottomed imaging dishes (micro-insert 4 well in µ-dish 35 mm, ibidi, Cat. 80406) coated with Matrigel as described above. Fluorescent time-lapse images of live myocultures were acquired on an Olympus IX83 inverted microscope equipped with a Yokogawa CSU-Wi spinning disc scanner unit using an UPlanXApo ×60/1.42 oil immersion objective. An Okolab UNO-T-H-$CO_2$ heated stage was used to carry on incubation with 5% $CO_2$ at 37°C during the entirety of time-lapse experiment. Tiled z-stacks were acquired every 15 min with a range of 10 µm at a step size of 1 µm, for up to 3 days. Closed culture dishes filled with ample media allowed for stable long-term imaging without perturbing the sample by media change. z-Drift compensation was used to keep a reference plane constant.

## Laser-ablation of live myofibers

Laser ablation and fluorescent live imaging of live myocultures were performed on a Nikon Eclipse TE 2000-E inverted spinning disc microscope (Ultraview ERS, Perkin Elmer) and a near-infrared (NIR, 1030 nm) femtosecond laser at 50 MHz repetition rate (t-Pulse, Amplitude Systems) (*Collinet et al., 2015*). Whole myofiber cuts were achieved by focusing the NIR laser using a ×100/1.4 oil immersion objective with an average power of 500 mW at the back aperture, then moving the focused beam along a 5–10 µm line perpendicular to the target myofiber at a speed of 300 µm/s using an xy-galvanometer (Cambridge Technologies). Fluorescent time-lapse images were acquired every 100 ms prior to, during, and post ablation for up to 10 s.

## Transcriptomics analysis

The bulk RNA-seq data analysis was performed on the already analysed data available under the GEO accession number GSE164874. Starting from the raw count matrix file 'GSE164874_SecondaryDifferentiation_raw_counts.xlsx', we kept here only the relevant conditions, namely the three replicates for the SKGM-2, KCTiP at 7 days and KCTiP at 15 days of differentiation. In order to compare each time point to a baseline level, we artificially generated three pseudo-replicates for the baseline condition. For each of the three pseudo-replicates, we computed a baseline value by averaging different replicates of the three time points. Raw count matrix filtered for the conditions of interest and with pseudo-replicates can be found in the *Supplementary file 1*. Normalisation, rlog transformation, and differential expression analysis were done in the R environment (v3.6.0), using the DESeq2 package (v 1.22.2) (*Love et al., 2014*). The output files of the DESeq2 analysis can be found in the *Supplementary file 1*.

Cluster analysis was performed on the normalised counts using the Mfuzz R package (v 2.42) (*Kumar and E Futschik, 2007*). To eliminate the noise, genes with lower than 50 total reads were filtered out. The data have been partitioned into 10 clusters according to the Dmin function provided by Mfuzz. Genes with membership values lower than 0.2 were filtered out generating non-overlapping clusters. Moreover, we performed 10 clustering runs and evaluated the stability of the clusters by computing the Jaccard Index. Cluster profiles are displayed in *Figure 3A*, while the gene membership lists in the clusters are available in *Supplementary file 1*. We performed GO enrichment of each cluster making use of Enrichr (*Kuleshov et al., 2016*). Results of the enrichment analysis can be found in the 'GO_enrichment_clusters.xlsx' supplementary file.

## Image analysis and quantification

### Segmenting and quantifying myofiber morphology, nuclei, adhesion, and maturation markers

Myofibers were manually segmented from ×10 phalloidin channels as freehand line ROIs in Fiji (*Schindelin et al., 2012*) by tracing with a stylus on a digital drawing pad. Length of these line ROIs were then measured in Fiji. Width of myofibers were manually quantified from ×100 phalloidin channels as length of line segment ROIs perpendicular to the myofibers in Fiji.

To count nuclei automatically, freehand line myofiber ROIs generated from ×10 phalloidin channels were reapplied to the DAPI channel to quantify pixel intensities along the line ROIs. A custom R script was developed to extract the numbers and positions of local maxima in DAPI pixel intensities above arbitrary thresholds as nuclei signal (*Dehapiot, 2022*). The results of automatic counting were compared with manual counting for selected images and the two were similar.

Quantifications of β1-integrin, titin, and actin levels for *Figure 8* were performed using line ROIs of a width of 50 pixels (4 μm) on multi-channel maximum projection images with a depth of 5 μm. Average pixel intensity per line ROI was then taken as a measure of relative protein levels.

### Segmenting and quantifying myofiber bundles and attachment foci

First, myofibers were manually segmented from ×10 phalloidin channels as freehand line ROIs in Fiji by tracing with a stylus on a digital drawing pad. Then, line ROIs were grouped as bundles with a custom script in R using a simple clustering algorithm based on the Hausdorff distance (*Borchers, 2021*). The Hausdorff distance is defined as the maximum distance of a set of points to their nearest point in the other set (*Rote, 1991*). The Hausdorff distance is chosen to quantify the distance between two line ROIs, as it takes into account every point and is thus superior than simply comparing average distances of line ROIs. Briefly, each set of line ROIs is represented by a set of xy coordinates. A Hausdorff distance is then calculated between each set of xy coordinates to obtain a distance matrix between pair-wise line ROIs. Line ROI pairs with Hausdorff distances below arbitrary thresholds are grouped into one myofiber cluster. One unique distance threshold is chosen for each time point, based on best matching between automatically generated clusters with visual inspections. Once segmented, the number of myofiber clusters is automatically counted. The area of each myofiber cluster is automatically quantified as the convex hull of all xy coordinates in line ROIs within that cluster.

Attachment foci are manually segmented from ×10 phalloidin channels, as sets of xy coordinates where one or more myofiber bundles converge.

### Vector field analysis of myofiber bundles

Vector field analysis of myofiber bundles was performed using the OrientationJ plugin (*Rezakhaniha et al., 2012*) in Fiji (*Schindelin et al., 2012*) combined with custom R scripts. First, in order to remove edges and empty space without myoculture, a central window of 3200×3200 pixels was cropped from the original 5000×5000 pixels images from ×10 phalloidin channels. Then, the cropped image was analysed using the OrientationJ vector field plugin with the following parameters: structure tensor local window σ=10 pixels, vector field grid size = 100 pixels. The output from OrientationJ (*Figure 2—figure supplement 1A–F*, *Figure 5—figure supplement 1A, B* ) is further analysed with custom scripts in R by adapting algorithms from *Guillamat et al., 2020*, as follows:

The spatial correlation C (*Figure 2—figure supplement 1G*) was defined as

$$C\left(d\right) = 2\,\left\langle \cos^2\left(\theta\left(\mathbf{r}\right) - \theta\left(\mathbf{r}+\mathbf{d}\right)\right)\right\rangle - 1$$

where θ(**r**) is the local orientation angle of the vector field at a grid position **r**, **d** is the separation vector to a second grid position **r+d** at distance |**d**|=d, and brackets ⟨ ⟩ denote an average over all pairs of grid points with same distance d. The characteristic nematic length $\eta_{\text{L}}$ was defined as the intersection of the initial linear decay of C(d) and the x axis (*Figure 2—figure supplement 1G*).

The local nematic order parameter S (*Figure 2—figure supplement 1A'–F'*, *Figure 5—figure supplement 1A, B*) was defined as

$$S\left(\text{x,y}\right) = \sqrt{\left\langle\cos\left(2\theta\right)\right\rangle^2_{(\text{x,y})} + \left\langle\sin\left(2\theta\right)\right\rangle^2_{(\text{x,y})}}$$

where θ is again the local orientation angle of the vector field at a given grid position, and brackets ⟨ ⟩ denote an average over a local regions of interest with center position (x,y). Specifically, we used square regions of interest comprising 4×4 grid positions each. The local nematic order parameter S was plotted as a heatmap (*Figure 2—figure supplement 1A'–F'*, *Figure 5—figure supplement 1A, B*). The fraction of disorder parameter is calculated as the fraction of local regions of interest for which the local nematic order parameter S is below a threshold of 0.5 (*Figure 2—figure supplement 1I*).

## Segmenting myofibrils and generating intensity profiles

1D ROIs of myofibrils were manually generated from 3D stacks of ×100 titin channel in four steps (*Figure 4D*): First, a spline is drawn on the xy plane of the original 3D stack. This xy-spline is used to generate a xy-straightened 3D volume (①). Second, the xy-straightened 3D volume is projected onto the xz plane, where a second spline is drawn. This xz spline is then combined with the xy spline to generate a xyz spline, which is then used to generate a straightened 3D volume (②). The straightened 3D volume is finally projected on the xy plane and its 1D intensity profile is extracted by averaging intensity values over the y axis (③). For subsequent automated 1D autocorrelation analysis (④, see section below for details).

## Autocorrelation analysis of sarcomere periodicity

Except for analysis reported in *Figures 5D and 6D–G*, autocorrelation analysis was performed using an automated procedure in Matlab (version 2019b) to detect periodicity in 1D titin intensity profiles (extracted from 1D ROIs, see previous section). The position and the prominence (amplitude) of the second correlation peak provides information about the spatial periodicity and the period regularity of the repetitive sarcomeric pattern (*Figure 4D*, panel ④).

For global autocorrelation analysis (*Figure 4E–F'*, *Figure 5C–E*), the ACF was calculated over the total length of the 1D profiles and averaged by culture day (*Figure 4E*) or by morphological categories (*Figure 4F*). For local autocorrelation analysis (*Figure 6*), the ACF was locally applied on 10 μm wide (~5 sarcomeres) sliding windows with a step size of 0.083 μm along the 1D profile. For each sliding window, the position and the prominence of the second autocorrelation peak were automatically extracted and the local portion of the fiber was classified as periodic if the prominence exceeded 0.6 (*Figure 6A*). Regions shorter than 6 μm were discarded as noise. Neighbouring regions less than 10 μm apart were connected. Four further quantities were calculated for each 1D titin intensity profiles: the proportion of periodic regions (*Figure 6B*), the average length of consecutive periodic regions (*Figure 6C*), the number of periodic regions per 100 μm of fiber (*Figure 6D*), and the homogeneity score calculated as the standard deviation of periodic window positions (*Figure 6E*). Autocorrelation analysis of 1D titin intensity profiles was performed in Matlab (version 2019b) with custom scripts (*Dehapiot, 2022*). Autocorrelation analysis detects periodicity from temporal or spatial series, and displays the period as the distance of the secondary peak from the y axis (*Figure 4D*, panel ④).

Autocorrelation analysis reported in *Figures 5D and 6D–G* was performed in R using the acf() function from the 'stats' package on manually segmented 1D intensity profile of fixed titin and live α-actinin-mKate2 signals at several time points matched by myofibril morphology. The prominence of the first ACF peak is automatically extracted from the resulting ACF profile (*Figures 5E, 6E and G*).

## Quantifying and modelling recoils of laser ablated myofibers

Pairs of post-ablation severed ends were tracked manually for a duration of 5 s from the point of ablation onwards, with the Manual Tracking plugin in FIJI (Fabrice P Cordelières, Institute Curie) and subsequently analysed in R with custom scripts (*Dehapiot, 2022*). Mean and standard deviation of distances between pairs of severed ends were plotted against time (*Figure 7B*, dots and shaded areas, respectively).

Two types of models were fitted to each recoil time course in R: First, a linear model was fitted on recoils during the first second. The initial recoil velocity was extracted as the slope of the linear model (*Figure 7B*, dashed line and C). Second, a viscoelastic model was fitted using an asymptotic regression model from the drc package (*Ritz et al., 2015*) with the following equation:

$$L(t) = L_{max} - (L_{max} - L_0) \cdot e^{-\frac{t}{\tau}}$$

where the recoil distance L is a function of time t, $L_{max}$ is the maximum attainable L, $L_0$ is L at time = 0, and $\tau$ (viscoelastic relaxation time) is inversely proportional to the relative rate of L increase while t increases (*Figure 7B*, solid curves and D). $\tau$ is defined by the ratio of the material viscosity to the Young's elastic modulus.

## Statistical analysis and data visualisation

Multiple pair-wise statistical tests in *Figures 6–8* were performed using the paired.t.test() function in R. Normality of datasets were verified with the Shapiro test using the shaprio.test() function. Statistical significance was reported as $p < 0.05$. With the exception to *Figure 4D–F'* and *Figure 6A*, all line and box plots were generated in R with standard plotting functions. Violin plots in *Figure 5* were generated with the 'vioplot' package (*Adler and Kelly, 2021*). *Figure 4D–F'* and *Figure 6A* were produced with Matlab, with *Figure 4E' and F'* and *Figure 6A* generated with the pseudo-colour plot (pcolor) function.

# Acknowledgements

We thank all members of the Schnorrer group, the Pourquié group, the IBDM, and the Centuri community for helpful discussions and feedback. We thank CE Seidman (Harvard University) and JG Seidman (Harvard University) for the gift of GFP-titin iPSC line, B Friedrich (TU Dresden) for valuable inputs on algorithms for vector field analysis, the IBDM imaging facility for microscopy support, and D Massey-Harroche (IBDM), E Bazellières (IBDM), V Dupuis (IBDM), and S Tlili (IBDM) for cell culture support. Funding: This work is supported by a Human Frontiers Science Program Grant to FS and OP (HFSP, RGP0052/2018). The Schnorrer lab is supported by the Centre National de la Recherche Scientifique (CNRS), the European Research Council under the European Union's Horizon 2020 Programme (ERC-2019-SyG 856118), the excellence initiative Aix-Marseille University A*MIDEX (ANR-11-IDEX-0001–02), the French National Research Agency with ANR-ACHN MUSCLE-FORCES and ANR-18-CE45-0016-01 MITO-DYNAMICS, the Bettencourt Foundation, the France-BioImaging national research infrastructure (ANR-10-INBS-04-01) and the Investissements d'Avenir, French Government program managed by the French National Research Agency (ANR-16-CONV-0001) and from Excellence Initiative of Aix-Marseille University – A*MIDEX (Turing Centre for Living Systems). MDC was funded by grant F31HD100033 from the Eunice Kennedy Shriver National Institute of Child Health and Human Development (NICHD) of the National Institutes of Health and the NASEM Ford Foundation Dissertation Fellowship. ARR has received funding from 'la Caixa' foundation (ID 100010434), under agreement LCF/BQ/AA18/11680032. The funders had no role in study design, data collection and analysis, decision to publish, or preparation of the manuscript.

# Additional information

### Funding

| Funder | Grant reference number | Author |
| --- | --- | --- |
| Human Frontier Science Program | RGP0052/2018 | Frank Schnorrer Olivier Pourquié |
| Centre National de la Recherche Scientifique | | Frank Schnorrer Pierre-François Lenne Bianca H Habermann |
| European Research Council | ERC-2019-SyG 856118 | Frank Schnorrer |
| Aix-Marseille Université | ANR-11-IDEX-0001-02 | Frank Schnorrer |
| Agence Nationale de la Recherche | MUSCLE-FORCES | Frank Schnorrer |
| Agence Nationale de la Recherche | ANR-18-CE45-0016-01 MITO-DYNAMICS | Frank Schnorrer Bianca H Habermann |

| Funder | Grant reference number | Author |
|---|---|---|
| Agence Nationale de la Recherche | ANR-10-INBS-04-01 | Pierre-François Lenne |
| Agence Nationale de la Recherche | ANR-16-CONV-0001 | Frank Schnorrer |
| Aix-Marseille Université | ANR-16-CONV-0001 | Frank Schnorrer |
| Turing Centre for Living Systems | ANR-16-CONV-0001 | Frank Schnorrer |
| Eunice Kennedy Shriver National Institute of Child Health and Human Development | F31HD100033 | Margarete Díaz-Cuadros |
| "la Caixa" Foundation | LCF/BQ/AA18/11680032 | Alejandra Rodríguez-delaRosa |

The funders had no role in study design, data collection and interpretation, or the decision to submit the work for publication.

## Author contributions

Qiyan Mao, Conceptualization, Data curation, Formal analysis, Validation, Investigation, Visualization, Methodology, Writing - original draft, Writing - review and editing; Achyuth Acharya, Data curation, Formal analysis, Validation, Investigation, Visualization, Methodology, Writing - original draft, Writing - review and editing; Alejandra Rodríguez-delaRosa, Investigation, Methodology, Writing - review and editing; Fabio Marchiano, Data curation, Formal analysis, Visualization, Methodology; Benoit Dehapiot, Software, Formal analysis, Visualization, Methodology, Writing - review and editing; Ziad Al Tanoury, Jyoti Rao, Margarete Díaz-Cuadros, Arian Mansur, Investigation, Methodology; Erica Wagner, Supervision, Investigation, Methodology; Claire Chardes, Data curation, Investigation, Methodology; Vandana Gupta, Resources, Supervision, Funding acquisition, Writing - review and editing; Pierre-François Lenne, Resources, Supervision, Funding acquisition, Methodology; Bianca H Habermann, Resources, Software, Supervision, Funding acquisition; Olivier Theodoly, Conceptualization, Supervision, Funding acquisition, Investigation, Methodology; Olivier Pourquié, Conceptualization, Resources, Supervision, Funding acquisition, Project administration, Writing - review and editing; Frank Schnorrer, Conceptualization, Resources, Formal analysis, Supervision, Funding acquisition, Visualization, Writing - original draft, Project administration, Writing - review and editing

## Author ORCIDs

Qiyan Mao ⓘ http://orcid.org/0000-0002-5564-0457
Benoit Dehapiot ⓘ http://orcid.org/0000-0002-7559-5497
Vandana Gupta ⓘ http://orcid.org/0000-0002-4057-8451
Pierre-François Lenne ⓘ http://orcid.org/0000-0003-1066-7506
Bianca H Habermann ⓘ http://orcid.org/0000-0002-2457-7504
Olivier Pourquié ⓘ http://orcid.org/0000-0001-5189-1227
Frank Schnorrer ⓘ http://orcid.org/0000-0002-9518-7263

## Decision letter and Author response

Decision letter https://doi.org/10.7554/eLife.76649.sa1
Author response https://doi.org/10.7554/eLife.76649.sa2

---

# Additional files

## Supplementary files

• Supplementary file 1. Raw and normalised RNAseq data of undifferentiated (SKGM), day 7 and day 15 myogenic cultures. GO-term enrichments of all 10 Mfuzz clusters.

• MDAR checklist

## Data availability

All data generated or analysed during this study are included in the manuscript and supporting file; Source Data files have been provided for all Figures; Table S1 contains the analysis of the sequencing data shown in Figure 3.

The following dataset was generated:

| Author(s) | Year | Dataset title | Dataset URL | Database and Identifier |
|---|---|---|---|---|
| Al Tanoury Z, Rao J, Marchiano F, Habermann B, Pourquié O | 2021 | GSE164874_SecondaryDifferentiation_raw_counts.xlsx | https://www.ncbi.nlm.nih.gov/geo/query/acc.cgi?acc=GSE164874 | NCBI Gene Expression Omnibus, GSE164874 |

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
