## [Editor Report]

This manuscript describes pioneering work providing a detailed description of iPS-derived muscle fiber differentiation in culture. It demonstrates that muscle fibers show self-organising capacities in vitro and form bundles with identified attachment points; this self-organisation generates internal tension within myofibers. Overall, this study suggests that tension drives sarcomerogenesis in multi-fibrillar vertebrate muscles and will be of interest to researchers in the muscle field and also biophysicists interested in collective cell behaviour.

---

## [Decision Letter]

**Decision letter after peer review:**

Thank you for submitting your article "Tension-driven multi-scale self-organisation in human iPSC-derived muscle fibers" for consideration by *eLife*. Your article has been reviewed by 3 peer reviewers, one of whom is a member of our Board of Reviewing Editors, and the evaluation has been overseen by Didier Stainier as the Senior Editor. The reviewers have opted to remain anonymous.

Essential revisions:

Overall the reviewers found the paper to be important, timely, well executed and beautifully presented. There is only one issue the reviewers would like the authors to try and provide more clarity about, hopefully with additional data. Specifically, the reviewers would like the authors to try and address the questions of first, whether bundling is necessary for myofibril formation and second whether fiber bundling increases attachment forces and thus tension inside the cells. As reviewer #2 notes "Previous reports, using primary muscle cells, showed rather isolated myofibers with myofibrils formed and functional. Is it due to the cells' origin or the culture method? Does myofibers' bundling increase attachment forces?". Reviewer #2 proposes some avenues to explore this which overlap with related comments by the other reviewers. Although the reviewers made suggestions that may extend the scope of the paper and strengthen the conclusions somewhat, these are not deemed essential for a revision.

*Reviewer #1 (Recommendations for the authors):*

– Laser microsurgery is indeed a powerful method to study tension but it is rather invasive. Can the authors vary tension using another approach (for example grow the cells on stretchable media, or vary cell density or media viscosity). I do not mean for a comprehensive study just a proof of principle.

–The behaviour, placement and positioning of nuclei is a topic of great interest in the field of muscle development. The authors have a powerful system to study these questions. The authors already use nuclei markers and track nuclei behaviour. Can they comment about nuclei behaviour and mobility during myofiber fusion, data which they already have access to?

– The authors use terms like "high tension" or "increasing tension"; these are not very precise terms. Measuring force is not trivial but the authors should give some thought and discuss how relative tensions are measured and give rise to different outcomes. Questions such "as is there a universal value that initiates developmental transitions or is the relative increase what triggers the change." should be explored.

*Reviewer #2 (Recommendations for the authors):*

The data presented here shows a solid temporal correlation between intracellular formation of myofibrils and maturation of myofibers, associated with the tension exerted on adhesion sites and inside the cells. I would like to make some comments and raise some questions that I hope will improve the article.

Authors describe that fibers coordinate themselves into bundles. However, is bundling necessary for myofibrils formation? Previous reports, using primary muscle cells, showed rather isolated myofibers with myofibrils formed and functional. Is it due to the cells' origin or the culture method? Does myofibers' bundling increase attachment forces? I think answering those questions will comfort the statement made in the article's title. Tension on substrate can be rather easily monitored using beads embedded in the matrix and follow their displacement over time.

If substrate stiffness is involved, and thus attachment forces, could the authors try to grow their cells on stiffer or softer substrate and monitor the appearance of periodic structures (sarcomeres)? Alternatively, cells could be grown on a deformable substrate (stretchable) which will give the authors the possibility to increase or reduce the tension forces between cell adhesion sites.

In the transcriptional analysis, it is rather surprising that clusters 8 and 9, which contain endoplasmic reticulum genes, appear to be downregulated during the differentiation process. The ER is becoming the SR in muscle cells and associates periodically with the sarcomeres; do not we expect a different outcome for the genes involved in ER biogenesis?

I am a bit concerned about the experiments using lasers to perform nano-lesions of the myofibrils, while in fact the whole cell is cut (cf methods). The observation of cell recoiling is thus a result of the ablation of both the membrane and the myofibrils and everything that was inside, the other cytoskeletons for instance. In a recent paper, Roman W et al. (Science 2021) have used a similar technique but only to sever a few myofibrils and not cut the whole cell. I suggest the authors use a similar approach and look at the sarcomeres on both sides of the damage to see how they initially contract towards the damage and if they go back to their initial state later, after myofibril repair.

Authors wrote that costameres are not present in their culture system. It is surprising and unlikely because myofibrillogenesis is dependent on the presence of costameres, which are providing a scaffold for sarcomere attachment. I suggest performing α-5-integrin, vinculin or FAK stainings to ascertain this.

In the discussion, authors wrote that "myotubes elongate during the initial culture phase via protrusive ends". While myoblast and myotube fusions can occur anywhere along a myotube, I do not understand the point made here. Myotubes mostly elongate by fusing with other cells at the initial phase; actually, the ratio between the number of nuclei inside a myotube and its area remain constant over time in this phase.

*Reviewer #3 (Recommendations for the authors):*

Specific comments:

1. Author's conclusions are based on temporal correlation between the appearance of sarcomeric pattern, clustering of myofiber's ends and an increase in tension and I am wondering whether there is a way to test this functionally. In Figure 5 they show that when one of the ends of mature myofiber detaches from foci sarcomeres collapse at the side of detachment. This however doesn't provide a functional link between tension and sarcomerogenesis as the detaching myofiber is already mature. It would be important to document whether sarcomeres form in the case detachment occurs earlier, at day 7 or so. Do authors see such kinds of events in their time-lapse experiments and/or could they by applying nano-lesions induce early detachment and then film detached myofiber in vivo?

2. Both Actin and B-integrin levels increase in differentiating myofibers and this is nicely documented, however I am less convinced by their accumulation at the myofiber ends with respect to the internal part of myofibers (as suggested in the scheme in Figure 8). This end-enrichment needs to be better illustrated and quantified.

---

## [Author Response]

Essential revisions:Overall the reviewers found the paper to be important, timely, well executed and beautifully presented. There is only one issue the reviewers would like the authors to try and provide more clarity about, hopefully with additional data. Specifically, the reviewers would like the authors to try and address the questions of first, whether bundling is necessary for myofibril formation and second whether fiber bundling increases attachment forces and thus tension inside the cells. As reviewer #2 notes "Previous reports, using primary muscle cells, showed rather isolated myofibers with myofibrils formed and functional. Is it due to the cells' origin or the culture method? Does myofibers' bundling increase attachment forces?". Reviewer #2 proposes some avenues to explore this which overlap with related comments by the other reviewers. Although the reviewers made suggestions that may extend the scope of the paper and strengthen the conclusions somewhat, these are not deemed essential for a revision.

We thank the reviewers for their thoughtful and constructive review of our manuscript. We were very happy to follow their advice throughout to further improve our manuscript and have now collected additional data further supporting the main conclusions of our manuscript.

Reviewer #1 (Recommendations for the authors):– Laser microsurgery is indeed a powerful method to study tension but it is rather invasive. Can the authors vary tension using another approach (for example grow the cells on stretchable media, or vary cell density or media viscosity). I do not mean for a comprehensive study just a proof of principle.

We agree with the reviewer that laser microsurgery is by definition invasive. We tried to minimise this effect by cutting with a very focused two-photon laser beam. This enabled us to only cut the myofibril without cutting the cell. These results showed that more mature myofibrils are under high tension (original Figure 6 —figure supplement 2, now Figure 7—figure supplement 2).

The additionally suggested experiments are all interesting to pursue. We do not have the cell-stretcher established yet. However, we have been able to plate the cells at varying densities (from 2.5K to 20K per cm^2^, formerly we used 10K for all experiments) and monitored their behaviour. We rationalised that bundling speed should be a readout of tension build-up and should be modified by varying cell density. Hence, we should see an effect on sarcomere assembly speed. In support of this hypothesis, we found that higher cell density indeed resulted in more efficient muscle fiber bundle formation, as analysed by unbiased vector field analysis that quantifies myofiber bundle alignment (new Figure 5 and Figure 5 —figure supplement 1). Importantly, our semi-automated autocorrelation analysis found that more efficient bundling in the denser cultures results in faster of assembly of their sarcomeres. At 10K and 20K seeding densities on day 5, we have seen significant advancement of sarcomere formation compared to 2.5K and 5K densities (new Figure 5C-E). At the 20K seeding density, we can even occasionally observe well-formed sarcomeres on day 3 (new Figure 5C). These new data further support the hypothesis that increased tension is responsible for efficient myofiber bundling and sarcomere assembly.

–The behaviour, placement and positioning of nuclei is a topic of great interest in the field of muscle development. The authors have a powerful system to study these questions. The authors already use nuclei markers and track nuclei behaviour. Can they comment about nuclei behaviour and mobility during myofiber fusion, data which they already have access to?

We agree with the reviewer that myonuclear behaviours are an exciting avenue of research in the future. However, we have not quantified nuclear movements in our live movies. We can see the nuclei in the phase contrast movies (not in the fluorescent spinning disc movies), but our current time resolution (30 min) does not always allow us to reliably track the nuclei between all the frames. Hence, we refrained from a detailed analysis of nuclear movement during or after fusion. Nuclear movements after fusion have been excellently quantified by the lab of Edgar Gomez using primary mouse myoblasts that more easily differentiate into mature fibers to quantify movement of the nuclei to the fiber surface after myofibrils have assembled (PMID 28892082). Our cultures are not yet reaching this level of maturity.

– The authors use terms like "high tension" or "increasing tension"; these are not very precise terms. Measuring force is not trivial but the authors should give some thought and discuss how relative tensions are measured and give rise to different outcomes. Questions such "as is there a universal value that initiates developmental transitions or is the relative increase what triggers the change." should be explored.

We thank the reviewer for this valuable suggestion. When revising the manuscript, we have tried to most accurately describe the tension values or tension changes over time and the possible impact of this change on for muscle morphogenesis. We hypothesise that it is rather a certain level of tension that needs to be reached to proceed to the next step of sarcomerogenesis. The evidence for this is that similarly looking myofibers have a comparable tension level. However, as we can measure tension only once per fiber and not over time, this is still a hypothesis, which we have extended in our discussion.

Reviewer #2 (Recommendations for the authors):The data presented here shows a solid temporal correlation between intracellular formation of myofibrils and maturation of myofibers, associated with the tension exerted on adhesion sites and inside the cells. I would like to make some comments and raise some questions that I hope will improve the article.Authors describe that fibers coordinate themselves into bundles. However, is bundling necessary for myofibrils formation? Previous reports, using primary muscle cells, showed rather isolated myofibers with myofibrils formed and functional. Is it due to the cells' origin or the culture method? Does myofibers' bundling increase attachment forces? I think answering those questions will comfort the statement made in the article's title. Tension on substrate can be rather easily monitored using beads embedded in the matrix and follow their displacement over time.

The reviewer is drawing attention to an important and critical point. Certainly, myofibrils can develop without the cells integrating into large bundles, even in our cell culture system. For the laser cutting experiments we have often chosen fibers that were only in small bundles to reduce the complexity of the imaging. So, tension is generated even if bundling is minimal. However, we are suggesting that bundling results in a higher build-up of tension, thus triggering sarcomerogenesis more effectively. This hypothesis is now further supported by new data gained from cultures with varying seeding densities. As explained above for reviewer 1, we have varied the seeding density from 2.5k to 20k cells per cm^2^, and found that denser seeding results in more efficient bundling and as a consequence faster sarcomere assembly (new Figure 5 and Figure 5 —figure supplement 1). This is consistent with the hypothesis that bundling promotes efficient sarcomere and myofibril assembly.

If substrate stiffness is involved, and thus attachment forces, could the authors try to grow their cells on stiffer or softer substrate and monitor the appearance of periodic structures (sarcomeres)? Alternatively, cells could be grown on a deformable substrate (stretchable) which will give the authors the possibility to increase or reduce the tension forces between cell adhesion sites.

These are both excellent suggestions by the reviewer, which are currently beyond our technical capacity. We plan to embark on these experiments in the near future to further probe the detailed mechanism of how substratum stiffness may impact upon tension and instruct bundling and sarcomere assembly.

In the transcriptional analysis, it is rather surprising that clusters 8 and 9, which contain endoplasmic reticulum genes, appear to be downregulated during the differentiation process. The ER is becoming the SR in muscle cells and associates periodically with the sarcomeres; do not we expect a different outcome for the genes involved in ER biogenesis?

Good point! We do not know why the ERAD pathway is down-regulated during maturation of the cultures. ERAD is targeting misfolded ER proteins for degradation. How well the SR has matured in our cultures has not yet been analysed. It is well possible that the expected up-regulation of SERCA, RyR and other core components of the SR is only happening at later stages that our cultures cannot reach yet (see below our response regarding costameres).

I am a bit concerned about the experiments using lasers to perform nano-lesions of the myofibrils, while in fact the whole cell is cut (cf methods). The observation of cell recoiling is thus a result of the ablation of both the membrane and the myofibrils and everything that was inside, the other cytoskeletons for instance. In a recent paper, Roman W et al. (Science 2021) have used a similar technique but only to sever a few myofibrils and not cut the whole cell. I suggest the authors use a similar approach and look at the sarcomeres on both sides of the damage to see how they initially contract towards the damage and if they go back to their initial state later, after myofibril repair.

The infrared two-photon laser that we have used here can indeed cut very precisely. Hence, we had already done the experiment the reviewer is favouring, it was presented in old Figure 6 —figure supplement 2 (now Figure 7 —figure supplement 2). With this method we only cut one or a part of an aligned myofibril bundle inside the cell and not the cell membrane. With this method we could document the fast recoil and hence large tension in the mature myofibrils but we did not investigate a possible repair. Such a repair can occur over a long duration (5 – 25 hours in Roman et al. Science 2021), thus it would be a major investment to look into this in detail, which was not our point with this experiment.

Authors wrote that costameres are not present in their culture system. It is surprising and unlikely because myofibrillogenesis is dependent on the presence of costameres, which are providing a scaffold for sarcomere attachment. I suggest performing α-5-integrin, vinculin or FAK stainings to ascertain this.

We indeed failed to detect clear costameres. We looked with anti-paxillin and antibeta1-integrin, and also with anti-vinculin and anti-alpha5-integrin (not shown). alpha5-integrin pairs with beta1, so we do not show the additional negative data. Our explanation is that our cultures are not yet mature enough to form proper, welldefined costameres recapitulating the sarcomeric spacings. We do find lots of integrin on the fiber surface, which we document. Regarding the assembly, we would argue the role of costameres is unclear. In insect flight muscle they are certainly not required, even mature flight muscles do not have costameres and the myofibrils purely span across the muscle cell from attachment to attachment (PMID 24631244). Here, we show that immature myofibrils assemble also across long distances, this argues that they are mechanically anchored throughout and not only locally at the next costameres. We argue they are mechanically anchored at the ends of the fibers or occasionally at lateral attachment sites, at which we do see large clusters of integrin-based attachments to the ECM and other fibers.

In the discussion, authors wrote that "myotubes elongate during the initial culture phase via protrusive ends". While myoblast and myotube fusions can occur anywhere along a myotube, I do not understand the point made here. Myotubes mostly elongate by fusing with other cells at the initial phase; actually, the ratio between the number of nuclei inside a myotube and its area remain constant over time in this phase.

We have documented the inverse relationship between myotube width and myotube length during the initial phase of myofiber elongation (Figure 1K, first 7 days). To explain this correlation, we proposed that myotubes elongate by one or two leading edges that move apart the cell centre (hence “extending”). This is supported by our live observations in phase contrast movies (Figure 1 —figure supplement 1B). We believe the new myoblasts largely fuse at the long axis of the cell and hence provide material that allow the leading edges to move further apart.

Reviewer #3 (Recommendations for the authors):Specific comments:1. Author's conclusions are based on temporal correlation between the appearance of sarcomeric pattern, clustering of myofiber's ends and an increase in tension and I am wondering whether there is a way to test this functionally. In Figure 5 they show that when one of the ends of mature myofiber detaches from foci sarcomeres collapse at the side of detachment. This however doesn't provide a functional link between tension and sarcomerogenesis as the detaching myofiber is already mature. It would be important to document whether sarcomeres form in the case detachment occurs earlier, at day 7 or so. Do authors see such kinds of events in their time-lapse experiments and/or could they by applying nano-lesions induce early detachment and then film detached myofiber in vivo?

We now provide new data that show that increased seeding density promotes more efficient bundling which promotes faster sarcomerogenesis (new Figure 5). On the one hand, we have never seen sarcomeres in detached fibers, it is hard to image detached fibers over time. They collapse, round up and swim away. However, most of the collapse indeed only occurs when tension is high after immature sarcomeres have assembled. On the other hand, long-term imaging after nano-lesions is an excellent suggestion. We are implementing this exact new experimental setup to investigate the consequence of tension release in a controlled manner in the near future.

2. Both Actin and B-integrin levels increase in differentiating myofibers and this is nicely documented, however I am less convinced by their accumulation at the myofiber ends with respect to the internal part of myofibers (as suggested in the scheme in Figure 8). This end-enrichment needs to be better illustrated and quantified.

The integrin end-enrichment is most obvious at the fiber-fiber contacts of large bundle ends. However, the reviewer is entirely correct that integrins are also found around the fibers in medial positions. The exact amounts are hard to compare as mature bundles are very dense, making quantitative imaging very tricky. As a compromise, we have added integrins also to the middle sections in our model in Figure 9 to better recapitulate these observations.